# Wasserstein Gradient Boosting: A Framework for Distribution-Valued Supervised Learning

**Takuo Matsubara**
The University of Edinburgh
Edinburgh, EH9 3JZ
takuo.matsubara@ed.ac.uk

## Abstract

Gradient boosting is a sequential ensemble method that fits a new weaker learner to pseudo residuals at each iteration. We propose Wasserstein gradient boosting, a novel extension of gradient boosting, which fits a new weak learner to alternative pseudo residuals that are Wasserstein gradients of loss functionals of probability distributions assigned at each input. It solves distribution-valued supervised learning, where the output values of the training dataset are probability distributions. In classification and regression, a model typically returns, for each input, a point estimate of a parameter of a noise distribution specified for a response variable, such as the class probability parameter of a categorical distribution specified for a response label. A main application of Wasserstein gradient boosting in this paper is tree-based evidential learning, which returns a distributional estimate of the response parameter for each input. We empirically demonstrate the competitive performance of the probabilistic prediction by Wasserstein gradient boosting in comparison with existing uncertainty quantification methods.

## 1 Introduction

Gradient boosting is a celebrated machine learning algorithm that has achieved considerable success with tabular data [1]. Gradient boosting has been extensively used for point forecasts and probabilistic classification, yet a relatively small number of studies have been concerned with the predictive uncertainty of gradient boosting. Predictive uncertainty of machine learning models plays a growing role in today's real-world production systems [2]. It is vital for safety-critical systems, such as medical diagnoses [3] and autonomous driving [4], to assess the potential risk of their actions that partially or entirely rely on predictions from their models. Gradient boosting has already been applied in a diverse range of real-world applications, including click prediction [5], ranking systems [6], scientific discovery [7], and data competition [8]. There is a pressing need for methodology to harness the power of gradient boosting to predictive uncertainty quantification.

In classification and regression, we typically specify a noise distribution $p(y \mid \theta)$ of a response variable $y$ and use a model to return a point estimate $\theta(x)$ of the response parameter for each input $x$. In recent years, the importance of capturing uncertainty in the model output $\theta(x)$ has increasingly been emphasised [2]. A variety of approaches have been proposed to obtain a distributional estimate $p(\theta \mid x)$ of the response parameter for each input $x$ [e.g. 9, 10, 11]. For example, Bayesian neural networks (BNNs) quantify uncertainty in network weights and propagate it to the space of network outputs. Marginalising the predictive distribution $p(y \mid \theta)$ over the distributional estimate $p(\theta \mid x)$ has been demonstrated to confer enhanced predictive accuracy and robustness against adversarial attacks [11]. Furthermore, the dispersion of the distributional estimate has been used as a powerful indicator for out-of-distribution (OOD) detection [12].

38th Conference on Neural Information Processing Systems (NeurIPS 2024).

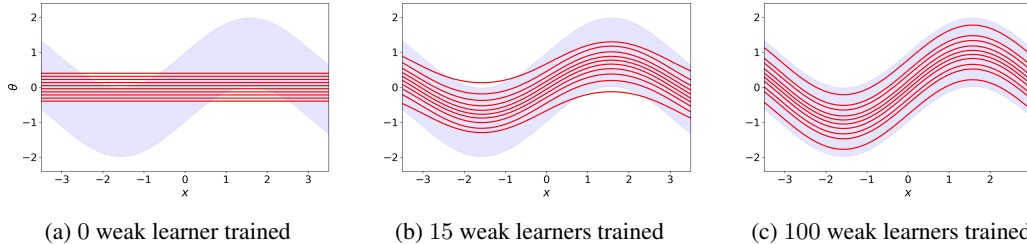

| (a) 0 weak learner trained | (b) 15 weak learners trained | (c) 100 weak learners trained |

Figure 1: Illustration of WGBoost trained on a set $\{x_i, \mu_i\}_{i=1}^{10}$ whose inputs are 10 grid points in $[-3.5, 3.5]$ and each output distribution is a normal distribution $\mu_i(\theta) = \mathcal{N}(\theta \mid \sin(x_i), 0.5)$ over $\theta \in \mathbb{R}$. The blue area indicates the 95% high probability region of the conditional distribution $\mathcal{N}(\theta \mid \sin(x), 0.5)$. WGBoost returns $N = 10$ particles (red lines) to predict the output distribution for each input $x$. This illustration uses the Gaussian kernel regressor for every weaker learner.

In this context, a line of research based on the concept of *evidential learning* has recently gained significant attention [11, 13, 14, 15]. The idea can be broadly interpreted as making use of the 'individual-level' posterior $p(\theta \mid y_i)$ of the response parameter $\theta$ conditional on each individual datum $y_i$, which arises from the response-distribution likelihood $p(y_i \mid \theta)$ and a user-specified prior $p(\theta)$. If each individual-level posterior falls into a closed form characterised by some hyperparameter, neural networks can be trained by using the finite-dimensional hyperparameter as a target value for each input. Outstanding performance and computational efficiency of the existing approaches have been delivered in a wide spectrum of engineering and medical applications [16, 17, 18, 19]. However, the existing approaches are limited to neural networks and to the case where every individual-level posterior is in closed form so that the finite-dimensional hyperparameter can be predicted by proxy. In general, posterior distributions are known only up to their normalising constants and, therefore, require an approximation typically by particles [20].

Without closed-form expression, each individual-level posterior needs to be treated as an infinite-dimensional output for each input. This challenge poses the following fundamental question:

> Consider a supervised learning setting whose outputs are probability distributions. Given a training set of input values and output distributions $\{x_i, \mu_i\}_{i=1}^D$, can we build a model that receives an input $x$ and returns a *nonparametric* prediction of the output distribution?

Motivated by this question, we propose a general framework of Wasserstein gradient boosting (WGBoost). WGBoost receives an input and returns a particle approximation of the output distribution. Figure 1 illustrates inputs and outputs of WGBoost. In this paper, we focus on application of WGBoost to evidential learning, where the individual-level posterior $p(\theta \mid y_i)$ of the response parameter $\theta$ is used as the output distribution $\mu_i$ for each input $x_i$ in the training set. Figure 2 compares the pipeline of evidential learning based on WGBoost with that of Bayesian learning.

**Contributions**  Our contributions are summarised as follows:

1. **Methodology of WGBoost**: Section 2 establishes the general framework of WGBoost. It is a novel family of gradient boosting that returns a set of particles that approximates an output distribution assigned at each input. In contrast to standard gradient boosting that fits a weak learner to the gradient of a loss function, WGBoost fits a weak learner to the estimated Wasserstein gradient of a loss functional over probability distributions.

2. **Application to Evidential Learning**: Section 3 establishes tree-based evidential learning based on WGBoost, with the loss functional specified by the Kullback–Leibler (KL) divergence. Following modern gradient-boosting libraries [21, 22] that uses second-order gradient boosting (c.f. Section 2.2), we implement a concrete second-order WGBoost algorithm built on an approximate Wasserstein gradient and Hessian of the KL divergence.

3. **Experiment on Real-world Data**: Section 4 demonstrates the performance of probabilistic regression, and classification with OOD detection, on real-world tabular datasets. To the author's knowledge, WGBoost is the first framework that enables evidential learning for (i) boosted tree models and (ii) cases without closed form of individual-level posteriors.

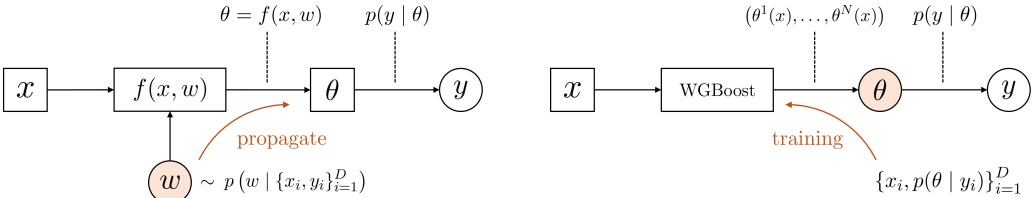

(a) Bayesian learning of a model $f(x, w)$     (b) Evidential learning based on WGBoost

Figure 2: Comparison of the pipeline of (a) Bayesian learning and (b) evidential learning based on WGBoost. The former uses the (global-level) posterior $p(w \mid \{x_i, y_i\}_{i=1}^{D})$ of the model parameter $w$ conditional on all data, and samples multiple models from it. The latter uses the individual-level posterior $p(\theta \mid y_i)$ of the response parameter $\theta$ as the output distribution of the training set, and trains WGBoost that returns a particle-based distributional estimate $p(\theta \mid x)$ of $\theta$ for each input $x$.

## 2   Wasserstein Gradient Boosting

This section establishes the general formulation of WGBoost. Section 2.1 recaps the notion of Wasserstein gradient flows, a 'gradient' system of probability distributions that minimises an objective functional in the space of probability distributions. Section 2.2 recaps the notion of gradient boosting, a sequential ensemble method that fits a new weak learner to the 'gradient' of the remaining loss at each iteration. Section 2.3 combines the above two notions to establish WGBoost, a novel family of gradient boosting that enables to solve distribution-valued supervised learning.

**Notation and Setting**   Let $\mathcal{X}$ and $\mathcal{Y}$ denote the space of inputs and responses in classification and regression. Suppose $\Theta = \mathbb{R}^d$. Let $\mathcal{P}_2$ be the 2-Wasserstein space i.e. a set of all probability distributions on $\Theta$ with finite second moment equipped with the Wasserstein metric [23]. We identify a probability distribution in $\mathcal{P}_2$ with its density whenever it exits. Denote by $\odot$ and $\oslash$, respectively, elementwise multiplication and elementwise division of two vectors in $\mathbb{R}^d$. Let $\nabla$ be the gradient operator. Let $\nabla_{\mathrm{d}}^2$ be a second-order gradient operator that takes the second derivative at each coordinate i.e. $\nabla_{\mathrm{d}}^2 f(\theta) = [\partial^2 f(\theta)/\partial \theta_1^2, \ldots, \partial^2 f(\theta)/\partial \theta_d^2]^\mathrm{T} \in \mathbb{R}^d$.

### 2.1   Wasserstein Gradient Flow

In the Euclidean space, a gradient flow of a function $f$ means a curve of points $x_t$ that solves a differential equation $(d/dt)x_t = -\nabla f(x_t)$ from some initial value $x_0$. That is the continuous-time limit of gradient descent, which minimises the function $f$ as $t \to \infty$. A Wasserstein gradient flow means a curve of probability distributions $\mu_t$ minimising a given functional $\mathcal{F}$ on the 2-Wasserstein space $\mathcal{P}_2$ from some initial distribution $\mu_0$. The Wasserstein gradient flow $\mu_t$ is characterised as a solution of a partial differential equation, known as the *continuity equation*:

$$\frac{d}{dt}\mu_t = -\nabla \cdot (\mu_t \nabla_W \mathcal{F}(\mu_t)) \quad \text{given} \quad \mu_0 \in \mathcal{P}_2, \tag{1}$$

where $\nabla_W \mathcal{F}(\mu) : \Theta \to \Theta$ denotes the *Wasserstein gradient* of $\mathcal{F}$ at $\mu$ [24, 25]. Appendix A recaps the derivation of the Wasserstein gradient, presenting the examples for several functionals.

One of the elegant properties of the Wasserstein gradient flow is casting the infinite-dimensional optimisation of the functional $\mathcal{F}$ as a finite-dimensional particle update [23]. The continuity equation (1) can be reformulated as a dynamical system of a random variable $\theta_t \sim \mu_t$, such that

$$\frac{d}{dt}\theta_t = -\left[\nabla_W \mathcal{F}(\mu_t)\right](\theta_t) \quad \text{given} \quad \theta_0 \sim \mu_0, \tag{2}$$

in the sense that the law $\mu_t$ of such a random variable $\theta_t$ is a weak solution of the continuity equation. Consider the case where the initial measure $\mu_0$ is set to the empirical distribution $\hat{\mu}_0$ of $N$ particles $\{\theta_0^n\}_{n=1}^{N}$. Discretising the continuous-time system (2) by the Euler method with a small step size $\nu > 0$ yields an iterative update scheme of $N$ particles $\{\theta_m^n\}_{n=1}^{N}$ from step $m = 0$:

$$\begin{bmatrix} \theta_{m+1}^1 \\ \vdots \\ \theta_{m+1}^N \end{bmatrix} = \begin{bmatrix} \theta_m^1 \\ \vdots \\ \theta_m^N \end{bmatrix} + \nu \begin{bmatrix} -[\nabla_W \mathcal{F}(\hat{\mu}_m)](\theta_m^1) \\ \vdots \\ -[\nabla_W \mathcal{F}(\hat{\mu}_m)](\theta_m^N) \end{bmatrix}, \tag{3}$$

where $\hat{\mu}_m$ denotes the empirical distribution of the particles $\{\theta_m^n\}_{n=1}^N$ at step $m$.

In practice, it is common that the Wasserstein gradient of a chosen functional $\mathcal{F}$ is not well-defined for empirical distributions. In such case, the particle update scheme (3) is not directly applicable because it depends on the Wasserstein gradient $\nabla_W \mathcal{F}(\hat{\mu}_m)$ at the empirical distribution $\hat{\mu}_m$. For example, the KL divergence $\mathcal{F}(\mu) = \mathrm{KL}(\mu \mid \pi)$ with a reference distribution $\pi$ leads to the Wasserstein gradient $[\nabla_W \mathcal{F}(\mu)](\theta) = -(\nabla \log \pi(\theta) - \nabla \log \mu(\theta))$ ill-defined if $\mu$ is an empirical distribution. Hence, the particle update scheme (3) is often performed with the estimated or approximated Wasserstein gradient well-defined for empirical distributions [e.g. 26, 27, 28, 29, 30]. A main application of WGBoost in Section 3 uses the 'smoothed' Wasserstein gradient of the KL divergence [26].

## 2.2 Gradient Boosting

Gradient boosting [31] is a sequential ensemble method of $M$ weak learners $f_1, \ldots, f_M$. It iteratively constructs an ensemble $F_m$ of $m$ weak learners $f_1, \ldots, f_m$ from step $m = 0$ to $M$. Given the current ensemble $F_m$ at step $m$, it trains a new weak learner $f_{m+1}$ to construct the next ensemble by

$$F_{m+1}(x) = F_m(x) + \nu f_{m+1}(x), \tag{4}$$

where $\nu$ is a shrinkage hyperparameter called a *learning rate*. The initial state of the ensemble $F_0(x)$ at step $m = 0$ is typically set to a constant that best fits the data. Any learning algorithm can be used as a weak learner in principle, although tree-based algorithms are most used [32].

The fundamental idea of gradient boosting is to train the new weak learner $f_{m+1}$ to approximate the negative gradient of the remaining error of the current ensemble $F_m$. Suppose that a loss function $L$ measures the remaining error $R_i(F_m(x_i)) := L(F_m(x_i), y_i)$ for each output vector $y_i \in \mathbb{R}^d$. The new weak learner $f_{m+1}$ is fitted to the set $\{x_i, g_i\}_{i=1}^D$ whose target variable $g_i$ is each specified by

$$g_i := -\nabla R_i(F_m(x_i)) \in \mathbb{R}^d.$$

The target $g_i$ is often called a *pseudo residual*. For each data input $x_i$, the boosting scheme (4) updates the output of the current ensemble $F_m(x_i)$ in the steepest descent direction of the error $R_i(F_m(x_i))$. Although [31] originally suggested an additional line search to determine a scaling constant of each weak learner, the line search has been reported to have a negligible influence on performance [33].

In modern gradient-boosting libraries, such as XGBoost [21] and LightGBM [22], the standard practice is to use the diagonal (coordinatewise) Newton direction of the remaining error $R_i(F_m(x_i))$ in lieu of the negative gradient $g_i$. The new base leaner $f_{m+1}$ is instead fitted to the set $\{x_i, g_i \oslash h_i\}_{i=1}^n$, where the negative gradient $g_i$ is divided elementwise by the Hessian diagonal $h_i$ given by

$$h_i := \nabla_d^2 R_i(F_m(x_i)) \in \mathbb{R}^d.$$

The target variable $g_i \oslash h_i$ is the diagonal Newton direction that minimises the second-order Taylor approximation of the remaining error $R_i(F_m(x_i))$ for each coordinate independently. Combining the second-order gradient boosting framework with tree-based weak learners has demonstrated exceptional scalability and performance [34, 35]. Although it is possible to use the 'full' Newton direction as the target variable of each weak learner, the impracticality of the full Newton direction has been pointed out [e.g. 36, 37]. In addition, the coordinatewise computability of the diagonal Newton direction is suitable for popular gradient-boosting tree algorithms [36].

## 2.3 General Formulation of Wasserstein Gradient Boosting

Now we consider the setting of distribution-valued supervised learning, where we are given a training set of input vectors and output distributions $\{x_i, \mu_i\}_{i=1}^D \subset \mathcal{X} \times \mathcal{P}_2$. Our goal is to construct a model that receives an input and returns a set of $N$ particles whose empirical distribution approximates the output distribution. We specify a loss functional $\mathrm{D}(\cdot \mid \cdot)$ between two probability distributions—such as the KL divergence—to measure the remaining error $\mathcal{F}_i(\cdot) = \mathrm{D}(\cdot \mid \mu_i)$ for each $i$-th training output distribution $\mu_i$. Our idea is to combine gradient boosting with Wasserstein gradient, where we iteratively construct a set of $N$ boosting ensembles $F_m^1, \ldots, F_m^N$ from step $m = 0$ to $M$.

Here, the output $F_m^n(x)$ of each $n$-th boosting ensemble represents the $n$-th output particle for an input $x$. Given the current set of $N$ ensembles $F_m^1, \ldots, F_m^N$ at step $m$, WGBoost trains a set of $N$

new weak learners $f_{m+1}^1, \ldots, f_{m+1}^N$ and computes the next set of $N$ ensembles by

$$
\begin{bmatrix} F_{m+1}^1(x) \\ \vdots \\ F_{m+1}^N(x) \end{bmatrix} = \begin{bmatrix} F_m^1(x) \\ \vdots \\ F_m^N(x) \end{bmatrix} + \nu \begin{bmatrix} f_{m+1}^1(x) \\ \vdots \\ f_{m+1}^N(x) \end{bmatrix} \tag{5}
$$

where $\nu$ is a learning rate. Similarly to standard gradient boosting, we specify the initial state of $N$ ensembles $F_0^1, \ldots, F_0^N$ at step $m = 0$ by a set of constants. Throughout, denote by $\hat{\mu}_{m,i}$ the empirical distribution of the $N$ output particles $F_m^1(x_i), \ldots, F_m^N(x_i)$ for each $i$-th training input $x_i$.

As discussed in Section 2.1, the Wasserstein gradient often needs to be estimated for empirical distributions. For better presentation, let $\mathcal{G}_i(\mu)$ denote an estimate of the Wasserstein gradient $\nabla_W \mathcal{F}_i(\mu)$ of the $i$-th remaining error $\mathcal{F}_i(\mu)$, which is well-defined for any distribution $\mu$. If the Wasserstein gradient $\nabla_W \mathcal{F}_i(\mu)$ is originally well-defined for any distribution $\mu$, it is a trivial choice of the estimate, i.e., $\mathcal{G}_i(\mu) = \nabla_W \mathcal{F}_i(\mu)$. Otherwise, any suitable estimate can be used as $\mathcal{G}_i(\mu)$. The foundamental idea of WGBoost is to train the $n$-th new learner $f_{m+1}^n$ to approximate the estimated Wasserstein gradient $-\mathcal{G}_i(\hat{\mu}_{m,i})$ evaluated at the $n$-th boosting output $F_m^n(x_i)$ for each $x_i$, so that,

$$
\begin{bmatrix} f_{m+1}^1(x_i) \\ \vdots \\ f_{m+1}^N(x_i) \end{bmatrix} \approx \begin{bmatrix} - \left[ \mathcal{G}_i(\hat{\mu}_{m,i}) \right] \left( F_m^1(x_i) \right) \\ \vdots \\ - \left[ \mathcal{G}_i(\hat{\mu}_{m,i}) \right] \left( F_m^N(x_i) \right) \end{bmatrix} .
$$

For each data input $x_i$, the boosting scheme (5) approximates the particle update scheme (3) for the output particles $F_m^1(x_i), \ldots, F_m^N(x_i)$ under the estimated Wasserstein gradient. The output particles are updated in the direction to decrease the remaining error $\mathcal{F}_i(\hat{\mu}_{m,i}) = \mathrm{D}(\hat{\mu}_{m,i} \mid \mu_i)$ at each step $m$.

Algorithm 1 summarises the general procedure of WGBoost. See Figure 1 for illustration of WGBoost. In Section 3, we choose the KL divergence as a loss functional D and use a kernel smoothing estimate of the Wasserstein gradient. See Appendix A for the Wasserstein gradient of other divergences.

**Remark 1** (**Stochastic WGBoost**). Stochastic gradient boosting [38] uses only a randomly sampled subset of data to fit a new weak learner at each step $m$ to reduce the computational cost. The same subsampling approach can be applied for WGBoost whenever the dataset is large.

**Remark 2** (**Second-Order WGBoost**). If any estimate of the Wasserstein 'Hessian' of the remaining error $\mathcal{F}_i$ is available, the Newton direction of $\mathcal{F}_i$ may also be computable [e.g. 39, 40]. Implemention of a second-order WGBoost algorithm is immediate by plugging such a Newton direction into $\mathcal{G}_i(\mu)$ in Algorithm 1. Our default WGBoost algorithm for tree-based evidential learning is built on a diagonal approximate Newton direction of the KL divergence, aligning with the standard practice in modern gradient-boosting libraries to use the diagonal Newton direction.

---

**Algorithm 1:** Wasserstein Gradient Boosting

---

**Input:** training set $\{x_i, \mu_i\}_{i=1}^D$ of input $x_i \in \mathcal{X}$ and output distribution $\mu_i \in \mathcal{P}_2$
**Parameter:** loss D, estimate $\mathcal{G}_i(\mu)$ of the Wasserstein gradient $\nabla_W \mathrm{D}(\mu \mid \mu_i)$, particle number
$\qquad\qquad N$, iteration $M$, learning rate $\nu$, weak learner $f$, initial constants $(\vartheta_0^1, \ldots, \vartheta_0^N)$
**Output:** set of $N$ boosting ensembles $(F_M^1, \ldots, F_M^N)$ at final step $M$
$(F_0^1(\cdot), \ldots, F_0^N(\cdot)) \leftarrow (\vartheta_0^1, \ldots, \vartheta_0^N)$ $\qquad\qquad$ ▷ set initial state of $N$ boosting ensembles
**for** $m \leftarrow 0, \ldots, M-1$ **do**
$\quad$ **for** $i \leftarrow 1, \ldots, D$ **do**
$\quad\quad$ $\hat{\mu}_{m,i} \leftarrow$ empirical distribution of output values $(F_m^1(x_i), \ldots, F_m^N(x_i))$ for input $x_i$
$\quad\quad$ **for** $n \leftarrow 1, \ldots, N$ **do**
$\quad\quad\quad$ $g_i^n \leftarrow - [\mathcal{G}_i(\hat{\mu}_{m,i})] (F_m^n(x_i))$ $\qquad$ ▷ compute target value of $n$-th new weak learner
$\quad\quad$ **end**
$\quad$ **end**
$\quad$ **for** $n \leftarrow 1, \ldots, N$ **do**
$\quad\quad$ $f_{m+1}^n \leftarrow \mathrm{fit}\left( \{x_i, g_i^n\}_{i=1}^D \right)$ $\qquad\qquad\qquad$ ▷ fit $n$-th new weak learner
$\quad\quad$ $F_{m+1}^n(\cdot) \leftarrow F_m^n(\cdot) + \nu f_{m+1}^n(\cdot)$ $\qquad$ ▷ set next state of $n$-th boosting ensemble
$\quad$ **end**
**end**

---

# 3 Application to Evidential Learning

This section provides our default setting to implement a concrete WGBoost algorithm for evidential learning, which enables classification and regression with predictive uncertainty. The individual-level posterior $p(\theta \mid y_i)$ of a response distribution $p(y \mid \theta)$ is used as the output distribution $\mu_i$ of the training set $\{x_i, \mu_i\}_{i=1}^{D}$. Section 3.1 recaps derivation of the individual-level posterior $p(\theta \mid y_i)$, followed by Section 3.2 discussing the default choice of the prior. We choose the KL divergence as a loss functional of WGBoost. Section 3.3 recaps a widely-used estimate of the Wasserstein gradient of the KL divergence based on kernel smoothing [26]. A further advantage of the kernel smoothing estimate is that the approximate Wasserstein Hessian is available, with which Section 3.4 establishes a second-order WGBoost algorithm similarly to modern gradient-boosting libraries.

## 3.1 Derivation of Individual-Level Posteriors and Predictive Distribution

Suppose that a response distribution $p(y \mid \theta)$ of a response variable $y$ is specified, as is typically done for probabilistic prediction. Suppose also that a prior distribution $p_i(\theta)$ of the response parameter $\theta$ is specified for each individual data input $x_i$. For each individual data pair $(x_i, y_i)$, the response-distribution likelihood $p(y_i \mid \theta)$ and the prior $p_i(\theta)$ determine the individual-level posterior

$$p(\theta \mid y_i) \propto p(y_i \mid \theta)p_i(\theta)$$

by Bayes' theorem. This individual-level posterior is set to the output distribution $\mu_i$ of the training set $\{x_i, \mu_i\}_{i=1}^{D}$ of WGBoost. The framework of WGBoost then constructs a model that returns a particle approximation of the output distribution $\mu_i(\cdot) = p(\cdot \mid y_i)$ for each data input $x_i$.

For a new input $x$, the constructed WGBoost model provides a set of particles $(\theta^1(x), \ldots, \theta^N(x))$ as a distributional prediction $p(\theta \mid x)$ of the response parameter $\theta$. We can define a predictive distribution $p(y \mid x)$ of the response $y$ for the new input $x$ via marginalisation of the output particles:

$$p(y \mid x) = \int_{\Theta} p(y \mid \theta)p(\theta \mid x)d\theta = \frac{1}{N}\sum_{i=1}^{N} p\left(y \mid \theta^i(x)\right). \tag{6}$$

We can also define a point prediction $\hat{y}$ for the new input $x$ via the individual-level Bayes action $\hat{y} = \operatorname{argmin}_{y \in \mathcal{Y}} \int_{\Theta} U(y, \theta)p(\theta \mid x)d\theta$, which minimises the average of some error $U : \mathcal{Y} \times \Theta \to \mathbb{R}$. For example, the Bayes action $\hat{y}$ is simply the mean of the output particles if $U(y, \theta) = (y - \theta)^2$.

In general, the explicit form of the individual-level posterior $p(\theta \mid y_i)$ is known only up to the normalising constant. Our full algorithm in Section 3.4 requires no normalising constant of the individual-level posterior $p(\theta \mid y_i)$. Our algorithm depends only on the log-gradient of the individual-level posterior $\nabla \log p(\theta \mid y_i)$ that cancels any constant term by the gradient. Hence, knowing the form of the response-distribution likelihood $p(y_i \mid \theta)$ and the prior $p_i(\theta)$ suffices.

**Remark 3 (Difference from Bayesian Learning).** Given a response distribution $p(y \mid \theta)$ and a model $\theta = f(x, w)$ with the parameter $w$, Bayesian learning of the model $f$ means the use of the posterior $p(w \mid \{x_i, y_i\}_{i=1}^{D})$ over $w$ conditional on all data. The predictive distribution $p(y \mid x)$ of the response $y$ is defined via marginalisation over $w$: $\int_{\Theta} p(y \mid \theta = f(x, w))p(w \mid \{x_i, y_i\}_{i=1}^{D})dw$. In contrast, WGBoost returns a distributional prediction $p(\theta \mid x)$ of the response parameter $\theta$, circumventing the marginalisation over the model parameter $w$ that can be ultra-high dimensional.

## 3.2 Choice of Individual-Level Priors

The prior distribution $p_i(\theta)$ of the response parameter $\theta$ is specified at each individual data input $x_i$. The approach to eliciting the prior may differ, depending on whether past data are available. If past data are available, they can be utilised to elicit a reasonable prior for unobserved data. If no past data are available, we recommend the use of a noninformative prior that have been developed as a sensible choice of prior in the absence of past data; see [e.g. 41] for the introduction. To avoid numerical errors, if a noninformative prior is improper (i.e. nonintegrable), we recommend the use of a proper probability distribution that approximates the noninformative prior sufficiently well.

**Example 1 (Normal Location-Scale Response).** Consider a scalar-valued response variable $y \in \mathbb{R}$ for regression. A normal location-scale response distribution $\mathcal{N}(y \mid m, \sigma)$ has the mean and scale parameters $m \in \mathbb{R}$ and $\sigma \in (0, \infty)$. A typical noninformative prior of $m$ and $\sigma$ are given by,

respectively, 1 and $1/\sigma$ which are improper. At every data point $(x_i, y_i)$, we use a normal prior $\mathcal{N}(m \mid 0, \sigma_0)$ over $m$ and an inverse gamma prior $\text{IG}(\sigma \mid \alpha_0, \beta_0)$ over $\sigma$, with the hyperparameters $\sigma_0 = 10$ and $\alpha_0 = \beta_0 = 0.01$, which approximate the non-informative priors.

**Example 2** (**Categorical Response**). Consider a label response variable $y \in \{1, \ldots, k\}$ for $k$-class classification. A categorical response distribution $\mathcal{C}(y \mid q)$ has the class probability parameter $q = (q_1, \ldots, q_k)$ in the $k$-dimensional simplex $\Delta_k$. If $k = 2$, it corresponds to the Bernoulli distribution. A typical noninformative prior of $q$ is given by $1/(q_1 \times \cdots \times q_k)$ which are improper. At every data point $(x_i, y_i)$, we use the logistic normal prior—a multivariate generalisation of the logit normal distribution [42]—over $q$ with the mean 0 and identity covariance matrix scaled by 10.

**Remark 4** (**Reparametrisation and Standardisation**). Section 2 supposed $\Theta = \mathbb{R}^d$ for some dimension $d$ without no loss of generality. Any parameter that lies in a subset of the Euclidean space (e.g. $\sigma$ in Example 1) can be reparametrised as one in the Euclidean space (e.g. $\log \sigma$). Appendix D details the reparametrisation used for Examples 1 and 2. In addition, if one's dataset has scalar outputs of a low or high order of magnitude, we recommend standardising the outputs.

### 3.3 Approximate Wasserstein Gradient of KL Divergence

We consider the KL divergence $\text{KL}(\mu \mid \mu_i)$ as a loss functional of WGBoost. One challenge of the KL divergence is that the resulting Wasserstein gradient $\left[\mathcal{G}_i^{\text{KL}}(\mu)\right](\theta) := -\left(\nabla \log \mu_i(\theta) - \nabla \log \mu(\theta)\right)$ is not well-defined when $\mu$ is an empirical distribution. A particularly successful solution—which originates in [43] and has been applied in wide contexts [26, 44, 45]—is to smooth the Wasserstein gradient through a kernel integral operator $\int_{\Theta} [\mathcal{G}_i^{\text{KL}}(\mu)](\theta^*)k(\theta, \theta^*)d\mu(\theta^*)$ [46]. By integration-by-part (see [e.g. 43]), the smoothed Wasserstein gradient, denoted $\mathcal{G}_i^*(\mu)$, falls into the following form that is well-defined for any distribution $\mu$:

$$[\mathcal{G}_i^*(\mu)](\theta) := -\mathbb{E}_{\theta^* \sim \mu}\left[\nabla \log \mu_i(\theta^*)k(\theta^*, \theta) + \nabla k(\theta^*, \theta)\right] \in \mathbb{R}^d, \tag{7}$$

where $\nabla k(\theta^*, \theta)$ denotes the gradient of $k$ with respect to the first argument $\theta^*$. An approximate Wasserstein gradient flow based on the smoothed Wasserstein gradient $\mathcal{G}_i^*(\mu)$ is called the Stein variational gradient descent [43] or kernelised Wasserstein gradient flow [47]. In most cases, the kernel $k$ is set to the Gaussian kernel $k(\theta, \theta^*) = \exp(-\|\theta - \theta^*\|^2/h)$ with the scale $h > 0$. Appendix B discusses a choice of kernel. This work uses the Gaussian kernel with $h = 0.1$ throughout.

Another common approach to approximating the Wasserstein gradient flow of the KL divergence is the Langevin diffusion approach [48]. The discretised algorithm, called the unadjusted Langevin algorithm [49], is a stochastic particle update scheme that adds a Gaussian noise at every iteration. However, several known challenges, such as asymptotic bias and slow convergence, often necessitate an ad-hoc adjustment of the algorithm [48]. Appendix B discusses a variant of WGBoost built on the Langevin algorithm, although it is not considered the default implementation.

### 3.4 Second-Order Implementation of WGBoost

We use a diagonal (coordinatewise) approximate Wasserstein Newton direction of the KL divergence, following the standard practice in modern gradient-boosting libraries [21, 22] to use the diagonal Newton direction of a loss. Similarly to smoothed Wasserstein gradient $\mathcal{G}_i^*(\mu)$, the approximate Wasserstein Hessian of the KL divergence $\text{KL}(\mu \mid \mu_i)$ can be obtained through the kernel smoothing. The diagonal of the approximate Wasserstein Hessian, denoted $\mathcal{H}_i^*(\mu)$, is defined by

$$[\mathcal{H}_i^*(\mu)](\theta) := \mathbb{E}_{\theta^* \sim \mu}\left[-\nabla_{\text{d}}^2 \log \mu_i(\theta^*)k(\theta, \theta^*)^2 + \nabla k(\theta, \theta^*) \odot \nabla k(\theta, \theta^*)\right] \in \mathbb{R}^d. \tag{8}$$

The diagonal approximate Wasserstein Newton direction of the KL divergence is then defined by $-[\mathcal{G}_i^*(\mu)](\cdot) \oslash [\mathcal{H}_i^*(\mu)](\cdot)$. Appendix C provides the derivation based on [39] who derived the Newton direction of the KL divergence in the context of nonparametric variational inference.

The second-order WGBoost algorithm is established by plugging it into $\mathcal{G}_i(\mu)$ in Algorithm 1, that is,

$$[\mathcal{G}_i(\mu)](\cdot) = [\mathcal{G}_i^*(\mu)](\cdot) \oslash [\mathcal{H}_i^*(\mu)](\cdot). \tag{9}$$

Algorithm 1 under the choice (9) is considered our default WGBoost algorithm for evidential learning. We refer this algorithm to as the *Wasserstein-boosted evidential learning* (WEvidential). The explicit pseudocode is provided in Algorithm 2 for full clarity.

**Algorithm 2:** Wasserstein-Boosted Evidential Learning

---

**Input:** dataset $\{x_i, y_i\}_{i=1}^D$ of input $x_i$ and response $y_i$ of classification or regression

**Parameter :** individual-level posterior $p(\theta \mid y_i)$ of response distribution $p(y \mid \theta)$, particle number $N$, iteration $M$, learning rate $\nu$, weak learner $f$, initial constants $\{\vartheta_0^n\}_{n=1}^N$

**Output:** set of $N$ boosting ensembles $(F_M^1, \ldots, F_M^N)$ at final step $M$

$(F_0^1(\cdot), \ldots, F_0^N(\cdot)) \leftarrow (\vartheta_0^1, \ldots, \vartheta_0^N)$  $\qquad\qquad\qquad$ ▷ set initial state of $N$ boostings

**for** $m \leftarrow 0, \ldots, M-1$ **do**
$\quad$ **for** $i \leftarrow 1, \ldots, D$ **do**
$\quad\quad$ $(\theta_i^1, \ldots, \theta_i^N) \leftarrow (F_m^1(x_i), \ldots, F_m^N(x_i))$  $\qquad$ ▷ get output particles for $i$-th data input
$\quad\quad$ **for** $n \leftarrow 1, \ldots, N$ **do**
$\quad\quad\quad$ $g_i^n \leftarrow \frac{1}{N} \sum_{k=1}^N \nabla \log p(\theta_i^k \mid y_i) k(\theta_i^k, \theta_i^n) + \nabla k(\theta_i^k, \theta_i^n)$
$\quad\quad\quad$ $h_i^n \leftarrow \frac{1}{N} \sum_{k=1}^N -\nabla_{\mathrm{d}}^2 \log p(\theta_i^k \mid y_i) k(\theta_i^k, \theta_i^n)^2 + \nabla k(\theta_i^k, \theta_i^n) \odot \nabla k(\theta_i^k, \theta_i^n)$
$\quad\quad$ **end**
$\quad$ **end**
$\quad$ **for** $n \leftarrow 1, \ldots, N$ **do**
$\quad\quad$ $f_{m+1}^n \leftarrow \mathrm{fit}\big( \{x_i, g_i^n \oslash h_i^n\}_{i=1}^D \big)$  $\qquad\qquad\qquad$ ▷ fit $n$-th new weak learner
$\quad\quad$ $F_{m+1}^n(\cdot) \leftarrow F_m^n(\cdot) + \nu f_{m+1}^n(\cdot)$  $\qquad\qquad\qquad$ ▷ set next state of $n$-th boosting
$\quad$ **end**
**end**

---

**Remark 5 (Computation).** The diagonal Newton direction (9) has the computational complexity $\mathcal{O}(N \times d)$ same as that of the smoothed Wasserstein gradient. Hence, there is essentially *no reason not to use* the diagonal Newton direction (9) instead of the smoothed Wasserstein gradient. Although it is possible to use the full Newton direction with no diagonal approximation, the computation requires the inverse and product of $(N \times d) \times (N \times d)$ matrices that result in the complexity up to $\mathcal{O}(N^3 \times d^3)$. Appendix D presents a simulation study to compare computational time and convergence speed of four WGBoost algorithms built on different estimates of the Wasserstein gradient.

## 4 Experiment on Real-world Tabular Data

We empirically demonstrate the performance of the WGBoost algorithm through three experiments using real-world tabular data. The first application illustrates the output of WGBoost through a simple conditional density estimation. The second application benchmarks the probabilistic regression performance. The third application demonstrates the classification and OOD detection performance. The source code is available in `https://github.com/takuomatsubara/WGBoost`.

**Common Hyperparameters** Throughout, we set the number of output particles $N$ to 10 and set each weak learner $f$ to the decision tree regressor [50] with maximum depth 1 for Section 4.1 and 3 for the rest. We set the learning rate $\nu$ to 0.1 for regression and 0.4 for classification, unless otherwise stated. Appendix E contains further details, including a choice of the initial constant $\{\vartheta_0^n\}_{n=1}^N$.

### 4.1 Illustrative Conditional Density Estimation

We illustrate the output of WEvidential by estimating a conditional density $p(y \mid x)$ from one-dimensional scalar inputs and outputs $\{x_i, y_i\}_{i=1}^D$. The normal output distribution $\mathcal{N}(y \mid m, \sigma)$ and the prior $p_i(m, \sigma)$ in Example 1 were used to define the individual-level posterior $p(m, \sigma \mid y_i)$, in which case the output of the WGBoost algorithm is a set of 10 particles $\{(m^n(x), \sigma^n(x))\}_{n=1}^{10}$ of the mean and scale parameters for each input $x$. We chose the number of weak learners $M$, drawing on an early-stopping approach used in [32], where we held out 20% of the training set as a validation set and chose the number $1 \leq M \leq 4000$ achieving the least validation error. Once the number $M$ was chosen, WEvidential was trained again using all the entire training set.

The conditional density is estimated using the predictive distribution (6) by WEvidential. We used two real-world datasets, *bone mineral density* [51] and *old faithful geyser* [52]. Figure 3 depicts the result for the former dataset, demonstrating that the WGBoost algorithm captures the heterogeneity of the conditional density on each input well. The result for the latter dataset is contained in Appendix E.1.

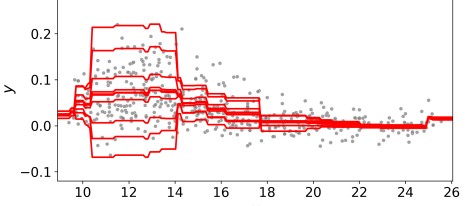 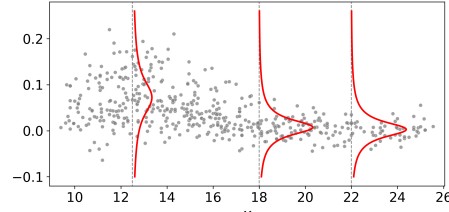

Figure 3: Conditional density estimation for the bone mineral density dataset (grey dots) by WEvidential, where the normal response distribution $\mathcal{N}(y \mid m, \sigma)$ is used for the response variable $y$. Left: distributional estimate (10 particles) of the location parameter $\{m^n(x)\}_{n=1}^{10}$ for each input. Right: estimated conditional density (6) through marginalisation of the output particles $\{(m^n(x), \sigma^n(x))\}_{n=1}^{10}$.

## 4.2 Probabilistic Regression Benchmark

We examine the regression performance of WEvidential using a standard benchmark protocol that originated in [53] and has been used in a number of subsequent works [10, 9, 32]. The benchmark protocol uses real-world tabular datasets from the UCI machine learning repository [54], each with one-dimensional scalar responses. As in Section 4.1, the normal response distribution $\mathcal{N}(y \mid m, \sigma)$ and the prior $p_i(m, \sigma)$ in Example 1 were used to define the individual-level posterior $p(m, \sigma \mid y_i)$.

We followed the data splitting protocol in [53] and randomly held out 10% of each dataset as a test set. The negative log likelihood (NLL) is measured by using the predictive distribution (6). The root mean squared error (RMSE) is measured by using the point prediction by the mean value of the predictive distribution. We chose the number of weak learners $M$ by the same approach as in Section 4.1. We repeated this procedure 20 times for each dataset, except the *protein* and *year msd* datasets for which we repeated five times and once. For the year msd dataset only, we subsampled 10% of data to fit each weak learner and used the learning rate 0.01 due to the large dataset size.

Table 1 compares the performance of WEvidential with five other methods: Monte Carlo Dropout (MCDropout) [9], Deep Ensemble (DEnsemble) [10], Concrete Dropout (CDropout) [55], Natural Gradient Boosting (NGBoost) [32], and Deep Evidential Regression (DEvidential) [13]. Appendix E provides further details on the experiment and a limited yet additional comparison. The WGBoost algorithm achieves the best score or a score sufficiently close to the best score most often.

Table 1: The NLLs and RMSEs for each dataset, where the best score is underlined and the scores whose standard deviation ranges include the best score are in bold. Results of MCDropout, DEnsembles, CDropout, NGBoost, and DEvidential were reported in [9], [10], [55], [32] and [13] respectively.

| Dataset | Criteria | WEvidential | MCDropout | DEnsemble | CDropout | NGBoost | DEvidential |
|---------|----------|-------------|-----------|-----------|----------|---------|-------------|
| boston | | **2.47 ± 0.16** | 2.46 ± 0.06 | **2.41 ± 0.25** | 2.72 ± 0.01 | **2.43 ± 0.15** | **2.35 ± 0.06** |
| concrete | | **2.83 ± 0.11** | 3.04 ± 0.02 | 3.06 ± 0.18 | 3.51 ± 0.00 | 3.04 ± 0.17 | 3.01 ± 0.02 |
| energy | | **0.53 ± 0.08** | 1.99 ± 0.02 | 1.38 ± 0.22 | 2.30 ± 0.00 | **0.60 ± 0.45** | 1.39 ± 0.06 |
| kin8nm | | -0.44 ± 0.03 | -0.95 ± 0.01 | -1.20 ± 0.02 | -0.65 ± 0.00 | -0.49 ± 0.02 | **-1.24 ± 0.01** |
| naval | NLL | -5.47 ± 0.03 | -3.80 ± 0.01 | -5.63 ± 0.05 | **-5.87 ± 0.05** | -5.34 ± 0.04 | -5.73 ± 0.07 |
| power | | **2.60 ± 0.04** | 2.80 ± 0.01 | 2.79 ± 0.04 | 2.75 ± 0.01 | 2.79 ± 0.11 | 2.81 ± 0.07 |
| protein | | 2.70 ± 0.01 | 2.89 ± 0.00 | 2.83 ± 0.02 | 2.81 ± 0.00 | 2.81 ± 0.03 | **2.63 ± 0.00** |
| wine | | **0.95 ± 0.08** | 0.93 ± 0.01 | **0.94 ± 0.12** | 1.70 ± 0.00 | **0.91 ± 0.06** | **0.89 ± 0.05** |
| yacht | | **0.16 ± 0.24** | 1.55 ± 0.03 | 1.18 ± 0.21 | 1.75 ± 0.00 | **0.20 ± 0.26** | 1.03 ± 0.19 |
| year msd | | 3.50 ± NA | 3.59 ± NA | **3.35 ± NA** | NA± NA | 3.43 ± NA | NA ± NA |
| boston | | **2.78 ± 0.60** | 2.97 ± 0.19 | **3.28 ± 1.00** | **2.65 ± 0.17** | **2.94 ± 0.53** | 3.06 ± 0.16 |
| concrete | | **4.15 ± 0.52** | 5.23 ± 0.12 | 6.03 ± 0.58 | 4.46 ± 0.16 | 5.06 ± 0.61 | 5.85 ± 0.15 |
| energy | | **0.42 ± 0.07** | 1.66 ± 0.04 | 2.09 ± 0.29 | 0.46 ± 0.02 | **0.46 ± 0.06** | 2.06 ± 0.10 |
| kin8nm | | 0.15 ± 0.00 | 0.10 ± 0.00 | 0.09 ± 0.00 | **0.07 ± 0.00** | 0.16 ± 0.00 | 0.09 ± 0.00 |
| naval | RMSE | **0.00 ± 0.00** | 0.01 ± 0.00 | **0.00 ± 0.00** | **0.00 ± 0.00** | **0.00 ± 0.00** | **0.00 ± 0.00** |
| power | | **3.19 ± 0.25** | 4.02 ± 0.04 | 4.11 ± 0.17 | 3.70 ± 0.04 | 3.79 ± 0.18 | 4.23 ± 0.09 |
| protein | | 4.09 ± 0.02 | 4.36 ± 0.01 | 4.71 ± 0.06 | **3.85 ± 0.02** | 4.33 ± 0.03 | 4.64 ± 0.03 |
| wine | | **0.61 ± 0.05** | **0.62 ± 0.01** | **0.64 ± 0.04** | 0.62 ± 0.00 | **0.63 ± 0.04** | **0.61 ± 0.02** |
| yacht | | **0.48 ± 0.18** | 1.11 ± 0.09 | 1.58 ± 0.48 | 0.57 ± 0.05 | **0.50 ± 0.20** | 1.57 ± 0.56 |
| year msd | | 9.11 ± NA | **8.85 ± NA** | 8.89 ± NA | NA± NA | 8.94 ± NA | NA ± NA |

Table 2: The classification accuracies and OOD detection PR-AUCs for each dataset, where the best score is underlined and in bold. The results other than WEvidential were reported in [14].

| Dataset | Criteria | WEvidential | MCDropout | DEnsemble | DDistillation | PNetwork |
|---|---|---|---|---|---|---|
| segment | Accuracy | $96.57 \pm 0.6$ | $95.25 \pm 0.1$ | **$97.27 \pm 0.1$** | $96.21 \pm 0.1$ | $96.92 \pm 0.1$ |
| | OOD | **$99.67 \pm 0.2$** | $43.11 \pm 0.6$ | $58.13 \pm 1.7$ | $35.83 \pm 0.4$ | $96.74 \pm 0.9$ |
| sensorless | Accuracy | **$99.54 \pm 0.1$** | $89.32 \pm 0.2$ | $99.37 \pm 0.0$ | $93.66 \pm 1.5$ | $99.52 \pm 0.0$ |
| | OOD | $81.13 \pm 5.3$ | $40.61 \pm 0.7$ | $50.62 \pm 0.1$ | $31.17 \pm 0.2$ | **$88.65 \pm 0.4$** |

### 4.3 Classification and Out-of-Distribution Detection

We examine the classification and anomaly OOD detection performance of WEvidential on two real-world tabular datasets, *segment* and *sensorless*, following the protocol used in [14]. The categorical response distribution $\mathcal{C}(y \mid q)$ and the prior $p_i(q)$ in Example 2 were used to define the individual-level posterior $p(q \mid y_i)$, in which case the output of the WGBoost algorithm is a set of 10 particles $\{q^n\}_{n=1}^{10}$ of the class probability parameter $q$ in the simplex $\Delta^k$ for each input $x$. We set the number of weak learners $M$ to $4000$ without early stopping to reduce the computational cost.

The segment and sensorless datasets have 7 and 11 classes in total. For the segment dataset, the data subset that belongs to the last class was kept as the OOD samples. For the sensorless dataset, the data subset that belongs to the last two classes was kept as the OOD samples. For each dataset, 20% of the non-OOD samples is held out as a test set to measure the classification accuracy. There exist several ways of defining a OOD score for each input [56]. For the WGBoost algorithm, the inverse of the maximum norm of the output-particle variance was used as the OOD score. We measured the OOD detection performance by the area under the precision recall curve (PR-AUC), viewing non-OOD test data as the positive class and OOD data as the negative class. We repeated this procedure five times.

Table 2 compares the performance of WEvidential with four other methods: MCDropout, DEnsemble, and Distributional Distillation (DDistillation) [57], and Posterior Network (PNetwork) [14]. Appendix E provides further details on the experiment. Figure 4 exemplifies how the dispersion of the output particles differ between OOD and non-OOD inputs. WEvidential demonstrates a high classification and OOD detection accuracy simultaneously. Although PNetwork has the best OOD detection performance for the sensorless dataset, the performance of the WGBoost algorithm also exceeds 80%, which is distinct from MCDropout, DEnsemble, and DDistillation.

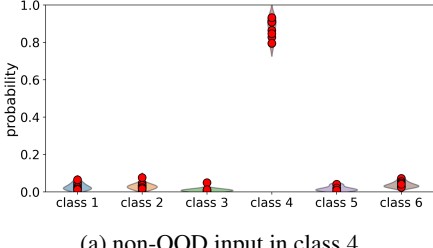
(a) non-OOD input in class 4

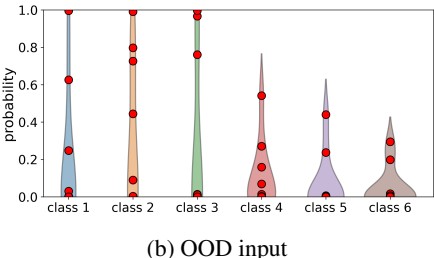
(b) OOD input

Figure 4: Examples of the output particles (red dot) of WEvidential on the segment dataset, where the coloured area indicate the kernel density estimation of the output particles for each class.

## 5 Discussion

This work established the general framework of WGBoost and developed the concrete algorithm WEvidential for evidential learning. The established framework of WGBoost offers exciting avenues for future research. Important directions for future study include (i) exploring alternative loss functionals to the KL divergence, (ii) investigating the convergence properties, and (iii) evaluating robustness of obtained predictive uncertainty in comparison to other methods. A particular limitation of WGBoost may arise when data are not tabular, as is the case of standard gradient boosting. These questions require careful examination and are critical for future study.

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

# Appendix

This appendix contains the technical and experiment details referred to in the main text. Appendix A recaps derivation of the Wasserstein gradient. Appendix B discusses the variant of WGBoost built on the unadjusted Langevin algorithm. Appendix C derives the diagonal approximate Wasserstein Newton direction used for WEvidential. Appendix D provides simulation studies on kernel choice of WEvidential and comparison of WGBoost algorithms built on different estimate of the Wasserstein gradient. Appendix E contains the additional details of the experiment presented in the main text.

## A   Derivation and Example of Wasserstein Gradient

This section recaps derivation of the Wasserstein gradient of a functional $\mathcal{F}$, with examples of common divergences. The Wasserstein gradient depends on a function on $\Theta$ called the *first variation* [24]. The first variation $\delta\mathcal{F}(\mu)/\delta\mu$ of the functional $\mathcal{F}$ at $\mu$ is a function on $\Theta$ that satisfies

$$\lim_{\epsilon \to 0^+} \frac{\mathcal{F}(\mu + \epsilon\nu) - \mathcal{F}(\mu)}{\epsilon} = \int_\Theta \frac{\delta\mathcal{F}(\mu)}{\delta\mu}(\theta)\nu(\theta)d\theta$$

for all signed measure $\nu$ s.t. $\mu + \epsilon\nu \in \mathcal{P}_2$ for all $\epsilon$ sufficiently small. The Wasserstein gradient $\nabla_W \mathcal{F}(\mu)$ of the functional $\mathcal{F}$ at $\mu$ is derived as the gradient of the first variation (see [e.g. 24]):

$$[\nabla_W \mathcal{F}(\mu)](\theta) := \nabla \frac{\delta\mathcal{F}(\mu)}{\delta\mu}(\theta).$$

It is common to suppose that the functional $\mathcal{F}$ consists of three energies, which are determined by functions $U : \mathbb{R} \to \mathbb{R}$, $V : \Theta \to \mathbb{R}$, and $W : \Theta \to \mathbb{R}$ respectively, such that

$$\mathcal{F}(\mu) = \underbrace{\int_\Theta U(\mu(\theta))d\theta}_{\text{internal energy}} + \underbrace{\int_\Theta V(\theta)\mu(\theta)d\theta}_{\text{potential energy}} + \underbrace{\frac{1}{2}\int_{\Theta\times\Theta} W(\theta - \theta')\mu(\theta)d\theta\mu(\theta')d\theta'}_{\text{interaction energy}}.$$

For a functional $\mathcal{F}$ that falls into the above form, the Wasserstein gradient is derived as

$$[\nabla_W \mathcal{F}(\mu)](\theta) = \nabla U'(\mu(\theta)) + \nabla V(\theta) + \int_\Theta \nabla W(\theta - \theta')\mu(\theta')d\theta'$$

where $U'$ is the derivative of $U : \mathbb{R} \to \mathbb{R}$ [23]. The KL divergence $\mathcal{F}(\mu) = \mathrm{KL}(\mu \mid \pi)$ of a distribution $\pi$ falls into the form with $U(x) = x\log x$, $V(\theta) = -\log\pi(\theta)$, and $W(\theta) = 0$, where

$$\mathrm{KL}(\mu \mid \pi) = \int_\Theta \log\mu(\theta)\mu(\theta)d\theta + \int_\Theta -\log\pi(\theta)\mu(\theta)d\theta.$$

Table 3 presents examples of Wasserstein gradients of common divergences $\mathcal{F}(\mu) = \mathrm{D}(\mu \mid \pi)$.

In the context of Bayesian inference, the KL divergence is particularly useful among many divergences. The Wasserstein gradient of the KL divergence requires no normalising constant of a posterior distribution $\pi$. This is because the Wasserstien gradient depends only on the log-gradient of the posterior $\nabla\log\pi(\theta) = \nabla\pi(\theta)/\pi(\theta)$ of the target $\pi$, in which case the normalising constant of the target $\pi$ is cancelled out by fraction. Hence, any posterior known only up to the normalising constant can be used as the target distribution $\pi$ in the Wasserstein gradient of the KL divergence.

Table 3: Wasserstein gradients of four divergences: the KL divergence [58], the chi-squared divergence [47], the alpha divergence [59], and the maximum mean discrepancy [60].

| Divergence $\mathcal{F}(\mu) = \mathrm{D}(\mu \mid \pi)$ | Wasserstein gradient $[\nabla_W \mathcal{F}(\mu)](\theta)$ |
|---|---|
| $\mathrm{KL}(\mu \mid \pi)$ | $-(\nabla\log\pi(\theta) - \nabla\log\mu(\theta))$ |
| $\mathrm{Chi}^2(\mu \mid \pi)$ | $2\nabla(\mu(\theta)/\pi(\theta))$ |
| $\mathrm{Alpha}(\mu \mid \pi)$ | $(\mu(\theta)/\pi(\theta))^{\alpha-1}\nabla(\mu(\theta)/\pi(\theta))$ |
| $\mathrm{MMD}(\mu \mid \pi)$ | $\int_\Theta \nabla k(\theta,\theta')\mu(\theta)d\theta - \int_\Theta \nabla k(\theta,\theta')\pi(\theta)d\theta$ |

# B    Langevin Gradient Boosting for KL Divergence

If a chosen functional $\mathcal{F}$ on $\mathcal{P}_2$ is the KL divergence $\mathcal{F}(\mu) = \mathrm{KL}(\mu \mid \pi)$ of a target distribution $\pi$, the continuity equation (1) admits an equivalent representation as the Fokker-Planck equation [61]:

$$\frac{d}{dt}\mu_t = \nabla \cdot (\mu_t \nabla \log \pi) + \Delta \mu_t \quad \text{given} \quad \mu_0 \in \mathcal{P}_2 \tag{10}$$

where $\Delta$ denotes the Laplacian operator. Recall that the original continuity equation (1) can be reformulated as the deterministic differential equation (2) of a random variable $\theta_t \sim \mu_t$. In contrast, the Fokker-Planck equation (10) can be reformulated as a stochastic differential equation of a random variable $\theta_t \sim \mu_t$, known as the overdamped Langevin dynamics [62]:

$$d\theta_t = \nabla \log \pi(\theta_t) dt + \sqrt{2} dB_t \quad \text{given} \quad \theta_0 \sim \mu_0, \tag{11}$$

where $B_t$ denotes a standard Brownian motion. Note that the deterministic system (2) in the case of the KL divergence and the above stochastic system (11) are equivalent at population level, in a sense that the law of the random variable $\theta_t$ in both the systems solves the two equivalent equations.

At the algorithmic level, however, discretisation of each system leads to different particle update schemes. Set the initial distribution $\mu_0$ in (11) to the empirical distribution $\hat{\mu}_0$ of $N$ initial particles $\{\theta_0^n\}_{n=1}^N$. Discretising the stochastic system (11) by the Euler-Maruyama method with a step size $\nu > 0$ yields a stochastic update scheme of particles $\{\theta_m^n\}_{n=1}^N$ from step $m = 0$:

$$\begin{bmatrix} \theta_{m+1}^1 \\ \vdots \\ \theta_{m+1}^N \end{bmatrix} = \begin{bmatrix} \theta_m^1 \\ \vdots \\ \theta_m^N \end{bmatrix} + \nu \begin{bmatrix} \nabla \log \pi(\theta_m^1) + \sqrt{2/\nu}\,\xi^1 \\ \vdots \\ \nabla \log \pi(\theta_m^N) + \sqrt{2/\nu}\,\xi^N \end{bmatrix},$$

where each $\xi^n$ denotes a realisation from a standard normal distribution on $\mathbb{R}^d$. The above updating scheme of each $n$-th particle is known as the unadjusted Langevin algorithm [49]. We can define a variant of WGBoost by replacing the term $\mathcal{G}_i(\mu)$ in Algorithm 1 with $\nabla \log \mu_i(\cdot) + \sqrt{2/\nu}\,\xi_i$ where $\mu_i$ is an output distribution at each $x_i$ and $\xi_i$ is a realisation from a standard normal distribution. The procedure is summarised in Algorithm 3, which we call Langevin gradient boosting (LGBoost).

---

**Algorithm 3:** Langevin Gradient Boosting

---

**Input:** training set $\{x_i, \mu_i\}_{i=1}^D$ of input $x_i \in \mathcal{X}$ and output distribution $\mu_i \in \mathcal{P}_2$
**Parameter :** particle number $N$, iteration $M$, rate $\nu$, weak learner $f$, initial constants
$\qquad\qquad (\vartheta_0^1, \ldots, \vartheta_0^N)$
**Output:** set of $N$ boosting ensembles $(F_M^1, \ldots, F_M^N)$ at final step $M$
$(F_0^1(\cdot), \ldots, F_0^N(\cdot)) \leftarrow (\vartheta_0^1, \ldots, \vartheta_0^N)$
**for** $m \leftarrow 0, \ldots, M-1$ **do**
    **for** $n \leftarrow 1, \ldots, N$ **do**
        **for** $i \leftarrow 1, \ldots, D$ **do**
            $g_i^n \leftarrow \nabla \log \mu_i(F_m^n(x_i)) + \sqrt{2/\nu}\,\xi_i^n \quad \text{where} \quad \xi_i^n \sim \mathcal{N}(0, I_d)$
        **end**
        $f_{m+1}^n \leftarrow \mathrm{fit}\left(\{x_i, g_i^n\}_{i=1}^D\right)$
        $F_{m+1}^n(\cdot) \leftarrow F_m^n(\cdot) + \nu f_{m+1}^n(\cdot)$
    **end**
**end**

---

# C    Derivation of Approximate Wasserstein Newton Direction

This section derives the diagonal approximate Wasserstein Newton direction based on the kernel smoothing. The approximate Wasserstein Newton direction of the KL divergence was derived in [39] under a different terminology—simply, the Newton direction—from a viewpoint of nonparametric variational inference. We place their result in the context of approximate Wasserstein gradient flows. Appendix C.1 shows the derivation of the smoothed Wasserstein gradient and Hessian. Appendix C.2 defines the Newton direction built upon the smoothed Wasserstein gradient and Hessian, following the derivation in [39]. Appendix C.3 derives the diagonal approximation of the Newton direction.

## C.1 Smoothed Wasserstein Gradient and Hessian

Consider the one-dimensional case $\Theta = \mathbb{R}$ for simplicity. For a map $T : \mathbb{R} \to \mathbb{R}$ and a distribution $\mu \in \mathcal{P}_2$, let $\mu_t$ be the pushforward of $\mu$ under the transform $\theta \mapsto \theta + tT(\theta)$ defined with a time-variable $t \in \mathbb{R}$. This means that $\mu_t$ is a distribution obtained by change-of-variable applied for $\mu$. The Wasserstein gradient of a functional $\mathcal{F}(\mu)$ can be associated with the time derivative $(d/dt)\mathcal{F}(\mu_t)$ [23]. In what follows, we focus on the KL divergence $\mathcal{F}(\mu) = \mathrm{KL}(\mu \mid \pi)$ as a loss functional. Under a condition $T \in L^2(\mu)$, the time derivative at $t = 0$ satisfies the following equality

$$\frac{d}{dt}\,\mathrm{KL}(\mu_t \mid \pi)\Big|_{t=0} = \int_\Theta T(\theta) \left[\mathcal{G}^{\mathrm{KL}}(\mu)\right](\theta)\,d\mu(\theta) = \left\langle T, \mathcal{G}^{\mathrm{KL}}(\mu)\right\rangle_{L^2(\mu)}, \tag{12}$$

where $\mathcal{G}^{\mathrm{KL}}(\mu)$ denotes the Wasserstein gradient of $\mathcal{F}(\mu) = \mathrm{KL}(\mu \mid \pi)$ with the target distribution $\pi$ made implicit. It gives an interpretation of the Wasserstein gradient as the steepest-descent direction because the decay of the KL divergence at $t = 0$ is maximised when $T = -\mathcal{G}^{\mathrm{KL}}(\mu)$.

The 'smoothed' Wasserstein gradient can be derived by restricting the transform map $T$ to a more regulated Hilbert space than $L^2(\mu)$. A reproducing kernel Hilbert space (RKHS) $H$ associated with a kernel function $k : \mathbb{R} \times \mathbb{R} \to \mathbb{R}$ is the most common choice of such a Hilbert space [e.g. 26]. An important property of the RKHS $H$ is that any function $f \in H$ satisfies the *reproducing property* $f(\theta) = \langle f(\cdot), k(\theta, \cdot)\rangle_H$ under the associated kernel $k$ and inner product $\langle \cdot, \cdot \rangle_H$ [63]. As discussed in [e.g. 46], applying the reproducing property in (12) under the condition $T \in H$ and exchanging the integral order, the time derivative satisfies an alternative equality as follows:

$$\frac{d}{dt}\,\mathrm{KL}(\mu_t \mid \pi)\Big|_{t=0} = \int_\Theta \left\langle T(\cdot), k(\theta, \cdot)\right\rangle_H \left[\mathcal{G}^{\mathrm{KL}}(\mu)\right](\theta)\,d\mu(\theta)$$

$$= \left\langle T(\cdot), \int_\Theta \left[\mathcal{G}^{\mathrm{KL}}(\mu)\right](\theta)k(\theta, \cdot)d\mu(\theta)\right\rangle_H = \left\langle T, \mathcal{G}^*(\mu)\right\rangle_H \tag{13}$$

where $[\mathcal{G}^*(\mu)](\cdot) := \int_\Theta [\mathcal{G}^{\mathrm{KL}}(\mu)](\theta)k(\theta, \cdot)d\mu(\theta)$ corresponds to the smoothed Wasserstein gradient used in the main text. The decay of the KL divergence at $t = 0$ is maximised by $T = -\mathcal{G}^*(\mu)$.

Similarly, the Wasserstein Hessian of the functional $\mathcal{F}(\mu)$ can be associated with the second time derivative $(d^2/dt^2)\mathcal{F}(\mu_t)$ [23]. As discussed in [e.g. 46], the Wasserstein Hessian of the KL divergence, denoted $\mathrm{Hess}(\mu)$, is an operator over functions $T \in L^2(\mu)$ that satisfies

$$\frac{d^2}{dt^2}\,\mathrm{KL}(\mu_t \mid \pi)\Big|_{t=0} = \left\langle T, \mathrm{Hess}(\mu)T\right\rangle_{L^2(\mu)}. \tag{14}$$

See [46] for the explicit form of the Wasserstein Hessian. In the same manner as the smoothed Wasserstein gradient, applying the reproducing property in (14) under the condition $T \in H$ and exchanging the integral order, the second time derivative satisfies an alternative equality as follows:

$$\frac{d^2}{dt^2}\,\mathrm{KL}(\mu_t \mid \pi)\Big|_{t=0} = \left\langle T(\star_1), \left\langle \left[\mathrm{Hess}^*(\mu)\right](\star_1, \star_2), T(\star_2)\right\rangle_H \right\rangle_H \tag{15}$$

where $[\mathrm{Hess}^*(\mu)](\star_1, \star_2) := \langle k(\star_1, \cdot), \mathrm{Hess}(\mu)k(\star_2, \cdot)\rangle_{L^2(\mu)}$ is the smoothed Wasserstein Hessian and the symbols $\star_1$ and $\star_2$ denote the variables to which each of the two inner products is taken.

In the multidimensional case $\Theta = \mathbb{R}^d$, the transport map $T$ is a vector-valued function $T : \mathbb{R}^d \to \mathbb{R}^d$, where a similar derivation can be repeated by replacing $L^2(\mu)$ and $H$ with the product space of $d$ independent copies of $L^2(\mu)$ and $H$. It follows from Proposition 1 and Theorem 1 in [39]—which derives the explicit form of (13) and (15) under their terminology, first and second variations—that the explicit form of the smoothed Wasserstein gradient and Hessian is given by

$$\left[\mathcal{G}^*(\mu)\right](\cdot) = \mathbb{E}_{\theta \sim \mu}\left[-\nabla \log \pi(\theta)k(\theta, \cdot) - \nabla k(\theta, \cdot)\right] \in \mathbb{R}^d,$$

$$\left[\mathrm{Hess}^*(\mu)\right](\star_1, \star_2) = \mathbb{E}_{\theta \sim \mu}\left[-\nabla^2 \log \pi(\theta)k(\theta, \star_1)k(\theta, \star_2) + \nabla k(\theta, \star_1) \otimes \nabla k(\theta, \star_2)\right] \in \mathbb{R}^{d \times d}$$

where $\nabla^2$ denotes an operator to take the Jacobian of the gradient—i.e., $\nabla^2 f(\theta)$ is the Hessian matrix of $f$ at $\theta$—and $\otimes$ denotes the outer product of two vectors. Note that both the smoothed Wasserstein gradient and Hessian are well-defined for any distribution $\mu$ including empirical distributions.

## C.2 Approximate Wasserstein Newton Direction

In the Euclidean space, the Newton direction of an objective function is a direction s.t. the second-order Taylor approximation of the function is minimised. Similarly, [39] characterised the Newton direction $T^* : \mathbb{R}^d \to \mathbb{R}^d$ of the KL divergence $\mathrm{KL}(\mu \mid \pi)$ as a solution of the following equation

$$\left\langle \left\langle [\mathrm{Hess}^*(\mu)](\star_1, \star_2), T^*(\star_2) \right\rangle_H + [\mathcal{G}^*(\mu)](\star_1), V(\star_1) \right\rangle_H = 0 \quad \text{for all} \quad V \in H.$$

Here $\Theta = \mathbb{R}^d$ and $H$ is the product space of $d$ independent copies of the RKHS of a kernel $k$. To obtain a closed-form solution, [39] supposed that the Newton direction $T^*$ can be expressed in a form $T^*(\cdot) = \sum_{i=1}^n W^n k(\cdot, \theta^n)$ dependent on a set of each particle $\theta^n \in \Theta$ and associated vector-valued coefficient $W^n \in \mathbb{R}^d$. Once the set of the particles is given, the set of the associated vector-valued coefficients is determined by solving the following simultaneous linear equation

$$\begin{bmatrix} \sum_{n=1}^N [\mathrm{Hess}^*(\mu)](\theta^1, \theta^n) \cdot W^n \\ \vdots \\ \sum_{n=1}^N [\mathrm{Hess}^*(\mu)](\theta^N, \theta^n) \cdot W^n \end{bmatrix} = \begin{bmatrix} -[\mathcal{G}^*(\mu)](\theta^1) \\ \vdots \\ -[\mathcal{G}^*(\mu)](\theta^N) \end{bmatrix}. \tag{16}$$

These equations (16) can be rewritten in a matrix form [64]. Let $K := N \times d$. Define a block matrix $\mathbf{H} \in \mathbb{R}^{K \times K}$ and a block vector $\mathbf{G} \in \mathbb{R}^K$ by the following partitioning

$$\mathbf{H} = \begin{pmatrix} \mathbf{H}_{11} & \cdots & \mathbf{H}_{1N} \\ \vdots & \ddots & \vdots \\ \mathbf{H}_{N1} & \cdots & \mathbf{H}_{NN} \end{pmatrix} \quad \text{and} \quad \mathbf{G} = \begin{pmatrix} \mathbf{G}_1 \\ \vdots \\ \mathbf{G}_N \end{pmatrix}$$

with each block specified as $\mathbf{H}_{ij} := [\mathrm{Hess}^*(\mu)](\theta^i, \theta^j) \in \mathbb{R}^{d \times d}$ and $\mathbf{G}_i := [\mathcal{G}^*(\mu)](\theta^i) \in \mathbb{R}^d$. Define a block matrix $\mathbf{K} \in \mathbb{R}^{K \times K}$ and a block vector $\mathbf{W} \in \mathbb{R}^K$ by the following partitioning

$$\mathbf{K} := \begin{pmatrix} \mathbf{K}_{11} & \cdots & \mathbf{K}_{1N} \\ \vdots & \ddots & \vdots \\ \mathbf{K}_{N1} & \cdots & \mathbf{K}_{NN} \end{pmatrix} \quad \text{and} \quad \mathbf{W} := \begin{pmatrix} W^1 \\ \vdots \\ W^N \end{pmatrix}$$

with each block of $\mathbf{K}$ specified as $\mathbf{K}_{ij} := \mathbf{I}_d \times k(\theta^i, \theta^j) \in \mathbb{R}^{d \times d}$, where $\mathbf{I}_d$ denotes the $d \times d$ identity matrix. Notice that $\mathbf{W}$ is a block vector that aligns the vector-valued coefficients $\{W^n\}_{n=1}^N$. Using these notations, the optimal coefficients that solve (16) is simply written as $\mathbf{W} = -\mathbf{H}^{-1}\mathbf{G}$ [64].

Given the optimal coefficients $\mathbf{W} = -\mathbf{H}^{-1}\mathbf{G}$, the Newton direction $T^*(\theta^n)$ evaluated at the given particle $\theta^n$ for each $n = 1, \ldots, N$ can be written in the following block vector form

$$\begin{pmatrix} T^*(\theta^1) \\ \vdots \\ T^*(\theta^N) \end{pmatrix} = - \begin{pmatrix} \mathbf{K}_{11} & \cdots & \mathbf{K}_{1N} \\ \vdots & \ddots & \vdots \\ \mathbf{K}_{N1} & \cdots & \mathbf{K}_{NN} \end{pmatrix} \begin{pmatrix} \mathbf{H}_{11} & \cdots & \mathbf{H}_{1N} \\ \vdots & \ddots & \vdots \\ \mathbf{H}_{N1} & \cdots & \mathbf{H}_{NN} \end{pmatrix}^{-1} \begin{pmatrix} \mathbf{G}_1 \\ \vdots \\ \mathbf{G}_N \end{pmatrix} \tag{17}$$

To distinguish from the standard Newton direction in the Euclidean space, we call (17) the approximate Wasserstein Newton direction. The approximate Wasserstein Newton direction yields a second-order particle update scheme. Suppose we have particles $\{\theta_m^n\}_{n=1}^N$ to be updated at each step $m$. At each step $m$, define the above matrices $\mathbf{H}$ and $\mathbf{G}$ with the empirical distribution $\mu = \hat{\pi}_m$ of the particles $\{\theta_m^n\}_{n=1}^N$. Replacing the Wasserstein gradient in the particle update scheme (3) by the approximate Wasserstein Newton direction (17) provides the second-order update scheme in [39].

## C.3 Diagonal Approximate Wasserstein Newton Direction

We derive the diagonal approximation of the approximate Wasserstein Newton direction, which we used for our second-order WGBoost algorithm. A few approximations of the approximate Wasserstein Newton direction were discussed in [39] for better performance of their particle algorithm. We derive the diagonal approximation so that no matrix product and inversion will be involved. Specifically, we replace the matrices $\mathbf{K}$ and $\mathbf{H}$ in (17) by the diagonal approximations $\hat{\mathbf{K}}$ and $\hat{\mathbf{H}}$, that is,

$$\hat{\mathbf{K}} = \begin{pmatrix} \mathbf{I}_d & \cdots & \mathbf{0} \\ \vdots & \ddots & \vdots \\ \mathbf{0} & \cdots & \mathbf{I}_d \end{pmatrix} \quad \text{and} \quad \hat{\mathbf{H}} = \begin{pmatrix} \mathbf{h}_{11} & \cdots & \mathbf{0} \\ \vdots & \ddots & \vdots \\ \mathbf{0} & \cdots & \mathbf{h}_{NN} \end{pmatrix},$$

where $\mathbf{K}_{nn} = \mathbf{I}_d \times k(\theta^n, \theta^n) = \mathbf{I}_d$ for the Gaussian kernel $k$ used in this work, and the matrix $\mathbf{h}_{nn} \in \mathbb{R}^{d \times d}$ denotes the diagonal approximation of the diagonal block $\mathbf{H}_{nn}$ of $\mathbf{H}$.

Recall that $\mathbf{H}_{nn} = [\mathrm{Hess}^*(\mu)](\theta^n, \theta^n)$. Denote by $\mathrm{Diag}(\mathbf{A})$ the diagonal of a square matrix $\mathbf{A}$. The diagonal approximation $\mathbf{h}_{nn}$ is a diagonal matrix whose diagonal is $\mathrm{Diag}(\mathbf{H}_{nn})$. We plug the diagonal approximations $\hat{\mathbf{K}}$ and $\hat{\mathbf{H}}$ in (17). It follows from inverse and multiplication properties of diagonal matrices that the approximate Wasserstein Newton direction turns into a form

$$\begin{pmatrix} T^*(\theta^1) \\ \vdots \\ T^*(\theta^N) \end{pmatrix} = - \begin{pmatrix} \mathbf{h}_{11} & \cdots & \mathbf{0} \\ \vdots & \ddots & \vdots \\ \mathbf{0} & \cdots & \mathbf{h}_{NN} \end{pmatrix}^{-1} \begin{pmatrix} \mathbf{G}_1 \\ \vdots \\ \mathbf{G}_N \end{pmatrix} = \begin{pmatrix} -\mathbf{G}_1 \oslash \mathrm{Diag}(\mathbf{H}_{11}) \\ \vdots \\ -\mathbf{G}_N \oslash \mathrm{Diag}(\mathbf{H}_{NN}) \end{pmatrix}. \quad (18)$$

At an arbitrary particle location $\theta$, denote by $[\mathcal{H}^*(\mu)](\theta)$ the diagonal of the smoothed Wasserstein Hessian $[\mathrm{Hess}^*(\mu)](\theta, \theta)$. It is straightforward to see that the diagonal can be written as

$$[\mathcal{H}^*(\mu)](\cdot) = \mathbb{E}_{\theta \sim \mu}\Big[ -\nabla_{\mathrm{d}}^2 \log \pi(\theta) k(\theta, \cdot)^2 + \nabla k(\theta, \cdot) \odot \nabla k(\theta, \cdot) \Big].$$

Notice that $\mathrm{Diag}(\mathbf{H}_{nn}) = [\mathcal{H}^*(\mu)](\theta^n)$ by definition. It therefore follows from the formula (18) with $\mathbf{G}_n = [\mathcal{G}^*(\mu)](\theta^n)$ and $\mathrm{Diag}(\mathbf{H}_{nn}) = [\mathcal{H}^*(\mu)](\theta^n)$ that the diagonal approximate Wasserstein Newton direction at an arbitrary particle location $\theta$ can be independently computed by

$$-[\mathcal{G}^*(\mu)](\theta) \oslash [\mathcal{H}^*(\mu)](\theta).$$

We used this direction in Section 3. In the main text, the diagonal approximate Wasserstein Newton direction is defined for each loss functional $\mathcal{F}_i(\cdot) = \mathrm{D}(\cdot \mid \mu_i)$, with $\pi = \mu_i$, using the smoothed Wasserstein gradient $\mathcal{G}_i^*(\mu)$ and the diagonal of the smoothed Wasserstein Hessian $\mathcal{H}_i^*(\mu)$ for each $i$-th output distribution $\mu_i$.

# D  Simulation Study for WEvidential

This section provides simulation studies on kernel choice of WEvidential and comparison of different estimate of the Wasserstein gradient to use in the WGBoost framework.

## D.1  Choice of Kernel

We used the kernel smoothing estimate of the Wasserstein gradient of the KL divergence in order to built WEvidential. The smoothed Wasserstein gradient originates in an approximate Wasserstein gradient flow called Stein variational gradient descent (SVGD) [43]. It is fairly common in SVGD in practice to use the Gaussian kernel $k(\theta, \theta^*) = \exp(-\|\theta - \theta^*\|^2/h)$ with the scale $h > 0$. For WEvidential, the scale $h$ may be viewed as a hyperparameter to choose. We recommend using the value of the scale s.t. $0.01 \le h \le 1.0$ in general. This work uses the value $h = 0.1$ throughout the experiments in the main text. One may opt for performing more advanced tuning of the scale $h$.

We provide a simulation study to examine sensitivity to the scale value. We prepared a synthetic dataset $\{x_i, \mu_i\}_{i=1}^D$ whose inputs are 200 gird points on the interval $[-3.5, 3.5]$ and output distributions are normal distributions $\mu_i(\theta) = \mathcal{N}(\theta \mid \sin(x_i), 0.5)$ conditional on each $x_i$. We used WEvidential with different kernel scales $h = 0.001, 0.01, 0.1, 1.0, 10, 100$ and fitted it to the synthetic dataset $\{x_i, \mu_i\}_{i=1}^D$. The decision tree regressor with the maximum depth 3 was used for each weak learner. The learning rate and the number of weak learners were set to 0.1 and 100. The initial constant $\{\vartheta^n\}_{n=1}^{10}$ of WEvidential was set to 10 grid points in the interval $[-10, 10]$.

We computed the output of WEvidential for 500 grid points in the interval $[-3.5, 3.5]$. We used the maximum mean discrepancy (MMD) [65] to measure the approximation error between the empirical distribution $\hat{\mu}_i$ of the output of WEvidential and the output distribution $\mu_i$:

$$\mathrm{MMD}^2(\hat{\mu}_i, \mu_i) = \mathbb{E}_{\theta \sim \hat{\mu}_i, \theta' \sim \hat{\mu}_i}[k_{\mathrm{D}}(\theta, \theta')] - 2\mathbb{E}_{\theta \sim \hat{\mu}_i, \theta' \sim \mu_i}[k_{\mathrm{D}}(\theta, \theta')] + \mathbb{E}_{\theta \sim \mu_i, \theta' \sim \mu_i}[k_{\mathrm{D}}(\theta, \theta')]$$

where $k_{\mathrm{D}}$ is a Gaussian kernel $k_{\mathrm{D}}(\theta, \theta') = \exp(-(\theta - \theta')^2/s)$ with the scale $s = 0.025$. The total approximation error was measured by the MMD averaged over all the inputs. Figure 5 shows the total approximation error of WEvidential for each scale value $h$ in the common log scale, together with examples of the output of WEvidential for different values of $h$. It demonstrates that the total error is minimised by $h = 0.1$ and stays in a relatively small value range for $0.01 \le h \le 1.0$.

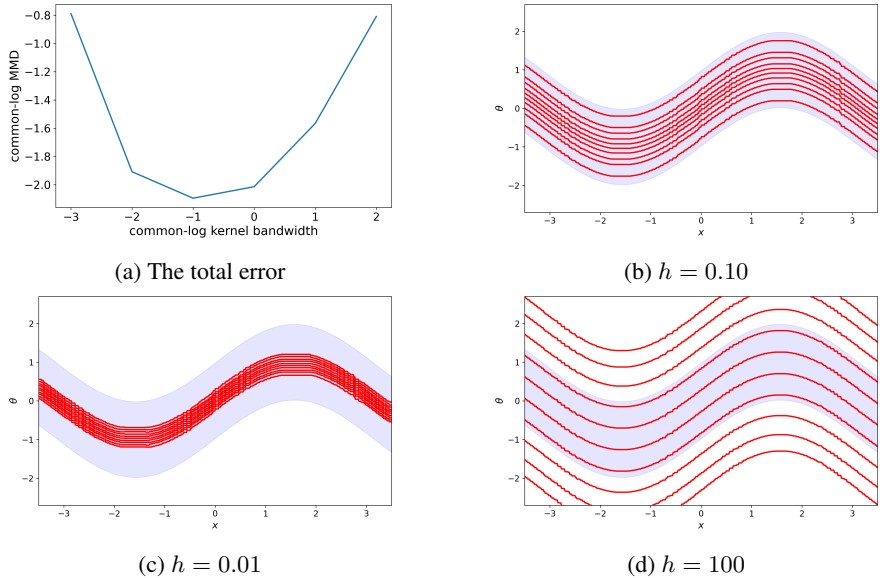

(a) The total error

(b) $h = 0.10$

(c) $h = 0.01$

(d) $h = 100$

Figure 5: The total MMD error and example outputs of WEvidential for different kernel scales. Panel (a): the total MMD error for different scale values $h = 0.001, 0.01, 0.1, 1.0, 10, 100$ both plotted in the common log scale. Panel (b): the output of WEvidential for $h = 0.1$. Panel (c): the output of WEvidential for $h = 0.01$. Panel (d): the output of WEvidential for $h = 100$.

### D.2 Comparison of Different Wasserstein Gradient Estimates

We compare four WGBoost algorithms built on different estimates of the Wasserstein gradient of the KL divergence. The first three algorithms set the estimate $\mathcal{G}_i(\mu)$ in Algorithm 1 to, respectively,

1. the smoothed Wasserstein gradient in (7);
2. the diagonal approximate Wasserstein Newton direction in (8);
3. the full approximate Wasserstein Newton direction in (17).

The fourth algorithm is LGBoost in Appendix B which is rather a variant of WGBoost. The first and third algorithms are called the first-order WEvidential and the full-Newton WEvidential, respectively. The second algorithm is our default WEvidential presented in Section 3. The first-order WEvidential is implemented by removing $h_i^n$ and replacing $g_i^n \oslash h_i^n$ with $g_i^n$ in Algorithm 2. The full-Newton WEvidential is implemented by replacing $g_i^n \oslash h_i^n$ in Algorithm 2 with $v_i^n$ computed by the following Algorithm 4, where $\nabla^2 f(\theta)$ denotes the Hessian matrix of a function $f : \Theta \to \mathbb{R}$ at $\theta$.

We prepared the same synthetic dataset $\{x_i, \mu_i\}_{i=1}^D$ as Appendix D.1 and fitted each algorithm to the dataset. We computed the output of each algorithm for 500 grid points in the interval $[-3.5, 3.5]$ and measured the approximation error by the MMD in the same manner as Appendix D.1. The decision tree regressor with the maximum depth 3 was used for each weak learner, and the learning rate was set to 0.1. We used an increasing number of weak leaners up to 100 weak learners, in order to observe the decay of the approximation error and the increase of the computational time. The initial constant $\{\vartheta^n\}_{n=1}^{10}$ for each algorithm was set to 10 grid points in the interval $[-10, 10]$, which sufficiently differs from the output distributions so that the decay of the approximation error is clear.

Figure 6 shows the approximation error and the computational time of each algorithm with respect to the increasing number of weak learners, together with the output of each algorithm with 100 weak learners trained. It demonstrates that WEvidential and full-Newton WEvidential reduce the approximation error most efficiently, while full-Newton WEvidential takes the longest computational time among others. As in Algorithm 4, the computation of the full approximate Wasserstein Newton direction requires the inverse and product of the $(N \times d) \times (N \times d)$ block matrices, where $N$ denotes the particle number $N$ and $d$ denotes the particle dimension. The error decay of LGBoost is not only slow but also shows stochasticity due to the Gaussian noise used in the algorithm. We therefore recommend our default WEvidential for better performance and efficient computation.

**Algorithm 4:** Computation of Full Approximate Wasserstein Newton Direction

---

**Input:** input $x_i$, output distribution $\mu_i$, outputs of $N$ boostings $(F_m^1(x_i), \ldots, F_m^N(x_i))$
**Output:** Wasserstein Newton direction $(v_i^1, \ldots, v_i^N)$ evaluated at $(F_m^1(x_i), \ldots, F_m^N(x_i))$

$(\theta_i^1, \ldots, \theta_i^N) \leftarrow (F_m^1(x_i), \ldots, F_m^N(x_i))$
**for** $n \leftarrow 1, \ldots, N$ **do**

$\quad g_i^n \leftarrow \frac{1}{N} \sum_{j=1}^N \nabla \log \mu_i(\theta_i^j) k(\theta_i^j, \theta_i^n) + \nabla k(\theta_i^j, \theta_i^n)$

$\quad$ **for** $k \leftarrow 1, \ldots, N$ **do**

$\quad\quad H_i^{nk} \leftarrow \frac{1}{N} \sum_{j=1}^N -\nabla^2 \log \mu_i(\theta_i^j) k(\theta_i^j, \theta_i^n) k(\theta_i^j, \theta_i^k) + \nabla k(\theta_i^j, \theta_i^n) \otimes \nabla k(\theta_i^j, \theta_i^k)$

$\quad\quad K_i^{nk} \leftarrow \mathbf{I}_d \cdot k(\theta_i^n, \theta_i^k)$

$\quad$ **end**

**end**

$$\begin{pmatrix} v_i^1 \\ \vdots \\ v_i^N \end{pmatrix} \leftarrow \begin{pmatrix} K_i^{11} & \cdots & K_i^{1N} \\ \vdots & \ddots & \vdots \\ K_i^{N1} & \cdots & K_i^{NN} \end{pmatrix} \begin{pmatrix} H_i^{11} & \cdots & H_i^{1N} \\ \vdots & \ddots & \vdots \\ H_i^{N1} & \cdots & H_i^{NN} \end{pmatrix}^{-1} \begin{pmatrix} g_i^1 \\ \vdots \\ g_i^N \end{pmatrix}$$

---

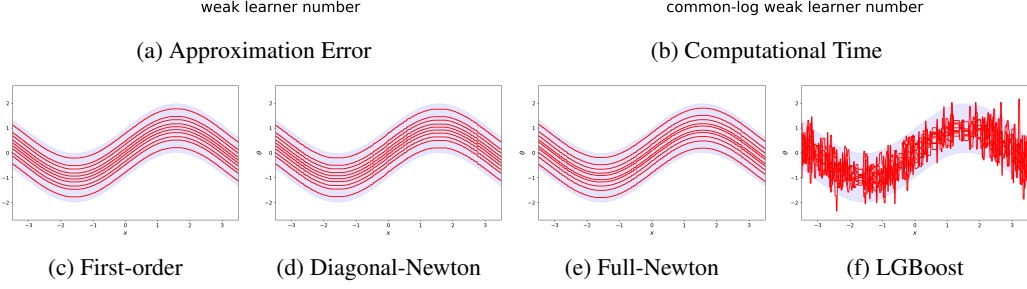

(a) Approximation Error

(b) Computational Time

(c) First-order  (d) Diagonal-Newton  (e) Full-Newton  (f) LGBoost

Figure 6: The approximation error and computational time of the four different WGBoost algorithms. Panel (a): the approximation error of each algorithm measured by the MMD averaged over the inputs with respect to the number of weak learners. Panel (b): the computational time with respect to the number of weak learners in common logarithm scale. Panel (c)-(f): the outputs of the four algorithms each with 100 weak learners used.

## E  Additional Details of Main Experiments

This section describes additional details of the applications in Section 4. All the experiments were performed with x86-64 CPUs, where some of them were parallelised up to 10 CPUs and the rest uses 1 CPU. The scripts to reproduce all the experiments are provided in the source code. Appendices E.1 to E.3 describe additional details of the applications in Sections 4.1 to 4.3 respectively. Appendix E.4 describes a choice of initial constants $\{\vartheta_0^n\}_{n=1}^N$ for the WGBoost algorithm used in Section 4.

### E.1  Detail of Section 4.1

The normal response distribution $\mathcal{N}(y \mid m, \sigma)$ in Example 1 was used in Section 4.1. The normal response distribution has the scale parameter $\sigma$ that lies only in the positive domain of the Euclidean space $\mathbb{R}$. We reparametrised it as one in the Euclidean space $\mathbb{R}$ by the log transform $\sigma' := \log(\sigma)$, which is the standard practice in Bayesian computation [20]. The inverse of the log transform

is the exponential transform $\sigma = \exp(\sigma')$. The change of variable formula tells the form of the individual-level posterior on $(m, \sigma')$ up to the normalising constant. Under the prior in Example 1, the individual-level posterior on $(m, \sigma')$ conditional on each individual response $y_i$ is given by

$$p(m, \sigma' \mid y_i) \propto \exp\left(-\frac{1}{2}\frac{(y_i - m)^2}{\exp(\sigma')^2}\right) \times \exp\left(-\frac{1}{2}\frac{m^2}{10^2}\right) \times \frac{1}{\exp(\sigma')^{1.01}}\exp\left(-\frac{0.01}{\exp(\sigma')}\right),$$

where the Jacobian determinant $|d\sigma/d\sigma'| = \exp(\sigma')$ is used for the change of variable. Figure 7 shows the output of WEvidential for the old faithful geyser dataset and the conditional density estimated through (6).

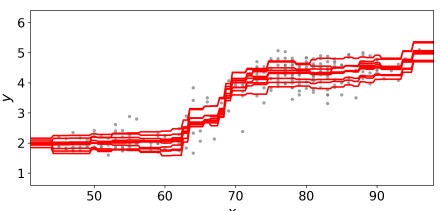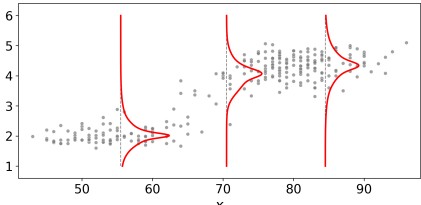

Figure 7: Conditional density estimation for the old faithful geyser dataset (grey dots) by WEvidential. Left: distributional estimate (10 particles) of the location parameter for each input. Right: estimated density by the predictive distribution (6) based on the output particles.

## E.2   Detail of Section 4.2

Section 4.2 used the same reparametrisation of the normal response parameter $(m, \sigma)$ as that of Appendix E.2. The NLL and RMSE of each algorithm on test data $\{x_i, y_i\}_{i=1}^D$ were computed by

$$\text{NLL} = -\frac{1}{D}\sum_{i=1}^D \log p(y_i \mid x_i) \quad \text{and} \quad \text{RMSE} = \sqrt{\frac{1}{D}\sum_{i=1}^D (y_i - \hat{y}_i)^2}$$

where $p(y_i \mid x_i)$ and $\hat{y}_i$ denote the predictive distribution and the point prediction provided by the algorithm. For WEvidential, we standardised the responses in the training data, in which case standardisation of test data were performed as $y_i' = (y_i - y_{\text{mean}}^{\text{train}})/y_{\text{std}}^{\text{train}}$ with the mean $y_{\text{mean}}^{\text{train}}$ and standard deviation $y_{\text{std}}^{\text{train}}$ of the training responses. Hence WEvidential provided the predictive distribution $p(y_i' \mid x_i)$ and the point prediction $\hat{y}_i'$ for the standardised responses $y_i'$. With no loss of generality, the NLL and RMSE for the original responses $y_i$ can be computed as follows:

$$\text{NLL} = -\frac{1}{D}\sum_{i=1}^D \log p(y_i \mid x_i) = -\frac{1}{D}\sum_{i=1}^D \log p(y_i' \mid x_i) + \log y_{\text{std}}^{\text{train}}, \tag{19}$$

$$\text{RMSE} = \sqrt{\frac{1}{D}\sum_{i=1}^D (y_i - (y_{\text{mean}}^{\text{train}} + y_{\text{std}}^{\text{train}} \times \hat{y}_i'))^2} = y_{\text{std}}^{\text{train}}\sqrt{\frac{1}{D}\sum_{i=1}^D (y_i' - \hat{y}_i')^2}, \tag{20}$$

where the equality of the NLL follows from the change of variable $p(y_i \mid x_i) = p(y_i' \mid x_i)/y_{\text{std}}^{\text{train}}$ and the equality of the RMSE follows from rearranging the terms.

We provide a brief description of each algorithm used in Section 4.2. For all the algorithms, the normal response distribution $p(y \mid m, \sigma)$ was specified as the response distribution, where the algorithm produces a point or distributional estimate of the response parameter $(m, \sigma)$ at each input $x$.

- MCDropout [9] trains a single neural network $F$ while dropping out each network weight with some Bernoulli probability. It can be interpreted as a variational approximation of a Bayesian neural network. It generates multiple subnetworks $\{F^n\}_{n=1}^N$ by subsampling the network weights by the dropout. The predictive distribution $p(y \mid x)$ is given by the model averaging $(1/N)\sum_{i=1}^N p(y \mid (m, \sigma) = F^n(x))$ for each input $x$.
- DEnsemble [10] simply trains independent copies $\{F^n\}_{n=1}^N$ of a neural network $F$ in parallel. It is one of the mainstream approaches to uncertainty quantification based on deep learning. The predictive distribution is given by the model averaging as in MCDropout.

- CDropout [55] consider a continuous relaxation of the Bernoulli random variable used in MCDropout to optimise the Bernoulli probability of the dropout. It generates multiple sub-networks $\{F^n\}_{n=1}^N$ by subsampling the network weights by the dropout with the optimised probability. The predictive distribution is the same as MCDropout.

- NGBoosting [32] is a family of gradient booting that use the natural gradient [66] of the response distribution as a target variable of each weak learner. In contrast to other methods, NGBoost outputs a single value $F(x)$ to be plugged into the response-distribution parameter. The predictive distribution $p(y \mid x)$ is given by $p(y \mid (m, \sigma) = F(x))$ for each input $x$.

- DEvidential [13] extends deep evidential learning [11], originally proposed in classification settings, to regression settings. It considers the case where the individual-level posterior of the response distribution falls into a conjugate parametric form, and predicts the hyperparameter of the individual-level posterior by a neural network. The predictive distribution is also given in a conjugate closed-form.

The algorithms used in Section 4.2 are computationally efficient uncertainty quantification methods that are commonly-used in practice. In the following, we provide a limited yet additional comparison of WEvidential with other Bayesian neural networks (BNNs) and a kernel-based model, Gaussian process (GP). We used a subset of the datasets in Section 4, which were used in [67] who employed BNNs with one hidden layer of 50 units. We additionally used a large-scale dataset, *keggd*. For the keggd dataset only, we reported the NLL and RMSE of the normalised outputs without the adjustment (19) and (20), in line with [67]. WEvidential was compared with BNNs learned by variational inference with the reparametrisation trick (VI), deterministic variational inference (DVI), stochastic weight averaging Gaussian (SWAG), principal component analysis subspace inference (PCA+VI), and two-layer deep Gaussian process with 50 induced points trained via expectation propagation (DGP1-50). Table 1 summarises the result.

Table 4: The NLLs and RMSEs for each dataset, where the best score is underlined. The results other than that of WEvidential were reported in [67].

| Dataset | Criteria | WEvidential | VI | DVI | SWAG | PCA+VI | DGP1-50 |
|---|---|---|---|---|---|---|---|
| boston | | 2.47 ± 0.16 | 2.43 ± 0.03 | 2.41 ± 0.02 | 2.76 ± 0.13 | 2.72 ± 0.13 | 2.33 ± 0.06 |
| concrete | | 2.83 ± 0.11 | 3.04 ± 0.02 | 3.06 ± 0.01 | 3.01 ± 0.09 | 2.99 ± 0.10 | 3.13 ± 0.03 |
| energy | NLL | 0.53 ± 0.08 | 2.38 ± 0.02 | 1.01 ± 0.06 | 1.70 ± 1.50 | 1.72 ± 1.59 | 1.32 ± 0.03 |
| naval | | -5.47 ± 0.03 | -5.87 ± 0.29 | -6.29 ± 0.04 | -6.71 ± 0.11 | -6.71 ± 0.11 | -3.60 ± 0.33 |
| yacht | | 0.16 ± 0.24 | 1.68 ± 0.04 | 0.47 ± 0.03 | 0.40 ± 0.42 | 0.40 ± 0.42 | 1.39 ± 0.14 |
| keggd | | -0.91 ± 0.04 | - | - | -1.08 ± 0.04 | -1.09 ± 0.03 | - |
| boston | | 2.78 ± 0.60 | - | - | 3.52 ± 0.98 | 3.46 ± 0.95 | - |
| concrete | | 4.15 ± 0.52 | - | - | 5.23 ± 0.42 | 5.14 ± 0.42 | - |
| energy | RMSE | 0.42 ± 0.07 | - | - | 1.59 ± 0.27 | 1.59 ± 0.27 | - |
| naval | | 0.00 ± 0.00 | - | - | 0.00 ± 0.00 | 0.00 ± 0.00 | - |
| yacht | | 0.48 ± 0.18 | - | - | 0.97 ± 0.38 | 0.97 ± 0.38 | - |
| keggd | | 0.24 ± 0.01 | - | - | 0.13 ± 0.03 | 0.13 ± 0.03 | - |

### E.3 Detail of Section 4.3

We reparametrised the parameter of the categorical response distribution $\mathcal{C}(y \mid q)$ used in Section 4.3, similarly to the normal response distribution used in Sections 4.1 and 4.2. The categorical response distribution has a class probability parameter $q = [q_1, \ldots, q_k]$ in the simplex $\Delta_k$. We applied the log-ratio transform $q' := [\log(q_1/q_k), \ldots, \log(q_{k-1}/q_k)] \in \mathbb{R}^{k-1}$ that maps from the simplex $\Delta_k$ to the Euclidean space $\mathbb{R}^{k-1}$ [42]. The inverse of the log-ratio transform is the logistic transform

$$q = \left[ \frac{\exp(q'_1)}{z_k}, \ldots, \frac{\exp(q'_{k-1})}{z_k}, \frac{1}{z_k} \right] \in \Delta_k \quad \text{where} \quad z_k = 1 + \sum_{j=1}^{k-1} \exp(q'_j).$$

The logistic normal distribution on the original parameter $q$ corresponds to a normal distribution on the transformed parameter $q'$ [42]. The change of variable formula tells the individual-level posterior on $q'$ up to the normalising constant. Under the prior in Example 2, the individual-level posterior

$p(q' \mid y_i)$ conditional on each individual observed response $y_i$ is given by

$$p(q' \mid y_i) \propto \left(\frac{1}{z_k}\right)^{[y_i=k]} \times \prod_{j=1}^{k-1} \left(\frac{\exp(q'_j)}{z_k}\right)^{[y_i=j]} \times \exp\left(-\frac{1}{2}\frac{\|q'\|^2}{10^2}\right),$$

where $[y_i = j]$ is $1$ if $y_i$ is the $j$-th class label and $0$ otherwise.

We provide a brief description of each algorithm used in Section 4.3. MCDropout and DEnsemble are described in Appendix E.2.

- DDistillation [57] predicts the parameter of a Dirichlet distribution over the simplex $\Delta_k$ by a neural network using the output of DEnsemble. The output of multiple networks in DEnsemble is distilled into the Dirichlet distribution controlled by one single network.

- PNetwork [14] considers the case where the individual-level posterior of the categorical response distribution falls into a Dirichlet distribution similarly to deep evidential learning [11]. It predicts the hyperparameter of the individual-level posterior given in the form of the Dirichlet distribution.

For the OOD detection, the OOD score we used was the inverse of the maximum norm of the variance of the WEvidential output. There are other quantities that are possible to use as the OOD score. For example, the entropy of the predictive distribution $p(y \mid x)$ is a quantity computable for any probabilistic classification method. We compared the OOD detection performance of WEvidential with that of NGBoost and Random Forest (RForest) based on the entropy of their predictive distributions. For reference, we also computed the OOD detection performance of WEvidential based on the entropy of the predictive distribution. For NGBoost, the decision tree regressor was used for each weak learner and 4000 weak learners were trained with the learning rate 0.01. For RForest, the maximum depth 3 was used. Table 5 shows the result on the segment dataset, demonstrating that the performance of WEvidential is higher, to a large margin, than that of NGBoost and RForest based on the entropy.

Table 5: The OOD detection performance of WEvidential, NGBoost, and RForest on the segment dataset, where WEvidential (Entropy) indicates the result of WEvidential based on the entropy.

| Dataset | Criteria | WEvidential | WEvidential (Entropy) | NGBoost | RForest |
|---|---|---|---|---|---|
| segment | Accuracy | 96.57 ± 0.6 | 96.57 ± 0.6 | 94.09 ± 0.6 | 88.18 ± 1.6 |
| | OOD | 99.67 ± 0.2 | 98.96 ± 0.4 | 89.96 ± 1.3 | 66.51 ± 1.1 |

To investigate on the effective learning rate of WEvidential for classification, we performed a simple simulation study of WEvidential using the synthetic *madelon* dataset. We created 600 data points of three-dimensional inputs and three-class labels. The data subset that belongs to the last class was kept as the OOD samples. We randomly held out 20% of non-OOD samples as a test set to measure the test classification accuracy. We trained WEvidential using different learning rates from 0.1 to 1.0 while fixing the number of weak learners to 4000. Table 6 shows that the result for the learning rate 0.4 demonstrates the best classification accuracy and the third best OOD detection performance. We also trained WEvidential using different numbers of weak learners from 1000 to 4000 while fixing the learning rate to 0.4. Table 7 shows that the result for the number of weak learners 4000 demonstrates the best classification accuracy and the second best OOD detection performance. While the most effective learning rate differs depending on each dataset, we drew on the insight obtained from the simulation study and set the learning rate to 0.4 for classification.

Table 6: The classification accuracy and OOD detection performance of WEvidential on the synthetic dataset for different learning rates, where the number of weak learners is fixed to 4000.

| Criteria | 0.1 | 0.2 | 0.3 | 0.4 | 0.5 | 0.6 | 0.7 | 0.8 | 0.9 | 1.0 |
|---|---|---|---|---|---|---|---|---|---|---|
| Accuracy | 92.50 | 92.50 | 92.50 | 93.75 | 91.25 | 86.25 | 90.00 | 75.00 | 85.00 | 80.00 |
| OOD | 29.85 | 34.79 | 35.26 | 41.77 | 40.52 | 42.69 | 35.21 | 35.58 | 42.04 | 36.05 |

Table 7: The classification accuracy and OOD detection performance of WEvidential on the synthetic dataset for different numbers of weak learners, where the learning rate is fixed to 0.4.

| Criteria | 1000 | 2000 | 3000 | 4000 |
|----------|-------|-------|-------|-------|
| Accuracy | 91.25 | 92.50 | 91.25 | 93.75 |
| OOD | 43.96 | 37.70 | 39.84 | 41.77 |

### E.4 Choice of Initial State of WGBoost

In standard gradient boosting, the initial state at step $m = 0$ is specified by a constant that most fits given data. Similarly, we specified the initial state $\{\vartheta_0^n\}_{n=1}^N$ of WEvidential by a set of constants that most fits the output distributions in average. We find such a set of constants by performing an approximate Wasserstein gradient flow averaged over all the output distributions. Specifically, given the term $\mathcal{G}_i(\mu)$ in Algorithm 1, we define $\bar{\mathcal{G}}(\mu) := (1/D) \sum_{i=1}^D \mathcal{G}_i(\mu)$ and perform the update scheme of a set of $N$ particles $\{\bar{\vartheta}_m^n\}_{n=1}^N$:

$$\begin{bmatrix} \bar{\vartheta}_{m+1}^1 \\ \vdots \\ \bar{\vartheta}_{m+1}^N \end{bmatrix} = \begin{bmatrix} \bar{\vartheta}_m^1 \\ \vdots \\ \bar{\vartheta}_m^N \end{bmatrix} + \nu_0 \begin{bmatrix} -[\bar{\mathcal{G}}(\hat{\pi}_m)](\bar{\vartheta}_m^1) \\ \vdots \\ -[\bar{\mathcal{G}}(\hat{\pi}_m)](\bar{\vartheta}_m^N) \end{bmatrix}$$

with the learning rate $\nu_0 = 0.01$ up to the maximum step number $m = 5000$. The initial particle locations for this update scheme were sampled from a standard normal distribution. We specified the initial state $\{\vartheta_0^n\}_{n=1}^N$ by the set of particles $\{\bar{\vartheta}_m^n\}_{n=1}^N$ obtained though this scheme at $m = 5000$.

