# OpenReview forum: "Wasserstein Gradient Boosting: A Framework for Distribution-Valued Supervised Learning"
_NeurIPS.cc/2024/Conference — NeurIPS 2024 poster_

### Official Review · Reviewer_DY4u · 2024-07-06

**Soundness:** 4
**Presentation:** 4
**Contribution:** 4
**Rating:** 7
**Confidence:** 3

**Summary:**

This work proposes to perform boosting with base learner which are fitted to the Wasserstein gradient of a loss function on the space of probability distributions, which can be useful in particular to capture uncertainty of the models.

Several variants of the algorithm are discussed (by adding a diagonal Hessian preconditioner and with different approximation of Wasserstein gradients for functionals not differentiable on discrete measures). The method is demonstrated on posterior regression tasks where the functionals are KL divergences with respect to some prior, and applied on different real datasets benchmarks.

**Strengths:**

This paper is well written and proposes a new interesting boosting method to minimize loss over probability distributions.

- The paper is well written
- A new boosting algorithm guided by functionals on probability distributions
- Application on real datasets outperforming baseline methods

**Weaknesses:**

The paper is good overall in my opinion, but still has some weaknesses.

- Experiments focus on KL divergence functional.
- No theoretical analysis

**Questions:**

Did you compare between the different algorithms (i.e. with and without Wasserstein Hessian preconditioner, and Kernel vs Langevin approximation) on the benchmark of Section 4.2?

Are there other possible functionals which could be interesting besides the KL or other divregences between probability distributions?

Could the method be used with input distributions, e.g. to do regression on probability distributions?

Typos:
- Line 145: "gradint"
- Line 520: "Wasserstien"

**Limitations:**

Yes

---

> ### Author Rebuttal · Authors · 2024-08-04
>
> Thank you for your strong endorsement of our work. We are delighted that you have found our methodology well-written and intriguing.
>
> &nbsp;
>
> > Did you compare between the different algorithms (i.e. with and without Wasserstein Hessian preconditioner, and Kernel vs Langevin approximation) on the benchmark of Section 4.2?
>
> Thank you for raising this point. So far we have compared them only through the simulation study to pick up the most reasonable algorithm (i.e. Wasserstein Hessian diagonal preconditioner) for the real-world application. We can add the further comparison of them based on the real-world application.
>
> &nbsp;
>
> > Are there other possible functionals which could be interesting besides the KL or other divregences between probability distributions?
>
> Thank you for checking this point with us. Yes, we have a large room to investigate what would be other interesting functionals to use with WGBoost. For example, kernel Stein discrepancy would be one of the examples that we can perform the Wasserstein gradient flow. We will expand the discussion about the other potential functional choices. Although our scope in this work is the KL divergence, it is important feature work to develop a further special case of WGBoost with other functionals.
>
> &nbsp;
>
> > Could the method be used with input distributions, e.g. to do regression on probability distributions?
>
> Thank you for your interesting question. Our current thought is that, because WGBoost can use any base learner, we can do that if we use such a base learner that can take a distribution as an input value. This sounds another interesting usage of WGBoost.
>
> &nbsp;
>
> > Typos: ...
>
> Thank you. We have corrected the pointed typos.

---

> > ### Comment · Reviewer_DY4u · 2024-08-09
> >
> > Thank you for your answer and addressing my comments. I do not have further questions and I will keep my score at 7.

---

> > > ### Author Response · Authors · 2024-08-09
> > > **Response to Reviewer**
> > >
> > > We appreciate your response and again all your efforts in reviewing our manuscript.

---

### Official Review · Reviewer_q9Jy · 2024-07-08

**Soundness:** 2
**Presentation:** 3
**Contribution:** 2
**Rating:** 3
**Confidence:** 5

**Summary:**

This paper proposed a new ensemble algorithm, called Wasserstein Gradient Boosting (WGBoost), which is a novel gradient boosting framework that leverages the wasserstein gradient for probabilistic prediction. Specifically, WGBoost fits a new base learner to the Wasserstein gradient of a loss functional on the space of probability distributions. Since we cannot access to the probability distribution at each iteration and we also cannot evaluate the wasserstein gradient at each iteration, the author proposed to implement it using the particle method and approximate the functional gradient by the kernel method. The algorithm returns a set of particles that approximate a target distribution for each input. The main application demonstrated is posterior regression, where WGBoost provides a distributional estimate of output-distribution parameters.

**Strengths:**

- WGBoost's ability to approximate target distributions with particles offers a robust approach to posterior regression,  capturing predictive uncertainty effectively.
- The proposed method shows superior performance in empirical evaluations on real-world tabular datasets, both for regression and out-of-distribution detection tasks.
- The implementation is seems easy by utilizing the particle method combined with kernel approximation.

**Weaknesses:**

The approach seems interesting, but the paper lacks the comparison with existing work. Moreover, there is no analysis or discussion when this algorithm is useful as I will explain in the below.
- It seems that the proposed method seems almost identical to the stein variational gradient descent (SVGD), but there is no qualitative and quantitative comparison with that. Please explain what is the fundamental difference compared to SVGD.
- Since the proposed method is very similar to SVGD, I think the comparison with SVGD and its extended methods including [3, 5, 6, 7] (I think there are other many variants of extention in SVGD).

- It has been known that the posterior approximation quality strongly depends on the choice of kernels in SVGD [1, 2, 3, 4]. Since the proposed algorithm is almost identical to SVGD, the quality of approximation by the WGBoost is strongly affected by the choice of the kernel function. However, there is no discussion about this point.

- The numerical experiments are only conducted with respect to the final performance on benchmark dataset and I cannot understand when and what kind of problems the proposed algorithm is suitable to  approximate the posterior distribution. It is known that SVGD suffers from collapse phenomena.
- In addition to the above point, there is no discussion about the computational cost of the proposed algorithm. When I say the computational cost, I point about the computational cost at each iteration and convergence speed. How large computational cost is compared to existing method regarding the number of particles and training dataset size ? I think the proposed algorithm is based on the boosting method, so it suffers from large computational cost with respect to the training dataset size.
- As for the convergence speed, it has been known that the convergence of the SVGD is slow, which does not show the linear convergence [4], and I suspect that the proposed method suffers similar problem. However, no discussion is present about the convergence speed or no numerical comparison exists with existing method.

[1] Stein Points

[2] Measuring Sample Quality with Kernels

[3] Kernel Stein Discrepancy Descent

[4] On the geometry of Stein variational gradient descent

[5] FUNCTION SPACE PARTICLE OPTIMIZATION FOR BAYESIAN NEURAL NETWORKS

[6] Feature Space Particle Inference for Neural Network Ensembles

[7] Repulsive Deep Ensembles are Bayesian

**Questions:**

I wrote questions in the Weakness.

**Limitations:**

The limitation is unclear. As far as I read no formal description is presented.

---

> ### Author Rebuttal · Authors · 2024-08-05
>
> Thank you for your assessment of our work. We are afraid that there seems a misinterpretation of our method in the reviewer's concerns: (a) the proposed method seems almost identical to the stein variational gradient descent (SVGD) and (b) the paper lacks the comparison with existing work. Please let us recap the high-level setting of WGBoost in the response that follows next. We hope this clarification will assist in the reassessment of our work and open up room for an upward revision of the score.
>
> &nbsp;
>
> > the proposed method seems almost identical to SVGD / the paper lacks the comparison with existing work
>
> **While SVGD is a sampling method, WGBoost is a new gradient boosting method**. Please note that our setting is **not** about sampling from a posterior over model parameters (e.g. BNN). To see this, we compare (i) learning of BNN and (ii) our setting here:
>
> - **BNN Learning**: We have a dataset $( x_i, y_i )\_{i=0}^{n}$ of input vector $x_i$ and output vector $y_i$, by which we have a posterior $P(w \mid ( x_i, y_i )\_{i=0}^{n} )$ over the network parameter $w$. The goal is to approximate $P(w \mid ( x_i, y_i )\_{i=0}^{n} )$ well by sampling or VI;
> - **Our Setting**: We have a dataset $( x_i, \pi_{x_i} )\_{i=0}^{n}$ where $\pi_{x_i}$ is a probability distribution given at each input $x_i$. Our goal is to have a learning algorithm that can predict the unseen distribution $\pi_x$ for any new input $x$.
>
> Our setting is clearly a less common, challenging regression problem because our target output is now a probability distribution $\pi_{x_i}$ at each input $x_i$ rather than some finite-dimensional vector output $y_i$. WGBoost is designed to solve this problem. Please find *Section B in our global rebuttal* for more detailed recap.
>
> Connection between WGBoost with SVGD is as follows. In Section 2, we described a general framework of WGBoost in Algorithm 1. To use WGBoost, we need to specify two components:
>
> - (a) a loss functional $\mathcal{F}(\hat{\pi}\_{x_i} | \pi\_{x_i})$ between the WGBoost output $\hat{\pi}\_{x_i}$ and the target distribution $\pi\_{x_i}$ at each $x_i$;
> - (b) how to estimate the Wasserstein gradient of $\mathcal{F}$ at each $x_i$.
>
> In Section 3.3, we described that the kernel-based functional gradient used in SVGD is one way to approximate the Wasserstein gradient of the KL divergence (see [49]). We then described that we use the kernel-based approximation to specify item (b) for our main applications. In Appendix E, we compared WGBoost under four different choices of the estimate of the Wasserstein gradient: functional gradient in SVGD, one in Stein Newton Method with full Hessian, one in Stein Newton Method with diagonal Hessian, and a stochastic alternative in Langevin diffusion. We applied WGBoost for probabilistic classification/regression. Hence our main experiment was the place to compare WGBoost with relevant UQ competitors.
>
> &nbsp;
>
> > ... the quality of approximation by the WGBoost is strongly affected by the choice of the kernel function. ...
>
> Thank you for raising this point. The choice of the kernel is indeed important for most kernel-related methods. We observed that the Gaussian kernel with a scale parameter 0.1 works well through a simulation study. We will add a detailed discussion on the kernel choice and the sensitivity with simulation studies.
>
> &nbsp;
>
> > The numerical experiments are only conducted with respect to the final performance on benchmark dataset and I cannot understand when and what kind of problems the proposed algorithm is suitable to approximate the posterior distribution.
>
> Thank you for the opportunity for us to clarify. Our proposed method is a classification/regression algorithm. For experiments using UCI data, reporting the total log-likelihood score or RMSE would be fairly a standard practice. For other simpler datasets, we have visualisation of the output of WGBoost: Figure 2 with the toy target with $\pi_{x_i} = \mathcal{N}(\sin(x_i), 1)$, Figure 1 for data in Section 4.1, and Figure 5 in Appendix for second data in Section 4.1. However, we agree that having more visualisation even for UCI datasets in Section 4.2-4.3 and adding a discussion on the approximation property would be helpful.
>
> For visualisation of UCI datasets, we will pick a few examples of input $x_i$ and target density $\pi_{x_i}$ from the dataset. We will then show the output of WGBoost at $x_i$ together with $\pi_{x_i}$. The input $x_i$ is of arbitrary dimension but the target density $\pi_{x_i}$ at each input $x_i$ is often 1 or 2 dimensional uni-modal density in our application. So the approximation at each $x_i$ should be less challenging. We will demonstrate this point through our additional visualisation.
>
> &nbsp;
>
> > In addition to the above point, there is no discussion about the computational cost of the proposed algorithm. ... / As for the convergence speed, it has been known that the convergence of the SVGD is slow ... I suspect that the proposed method suffers similar problem. However, no discussion is present about the convergence speed or no numerical comparison ...
>
> Thank you for the opportunity for us to clarify. As recapped at the top, WGBoost is not a sampling method such as SVGD. WGBoost is an algorithm that depends on user-specification of how to estimate the Wasserstein gradient. In Appendix E, we have a simulation study to compare both the loss-decay speed and computation time of WGBoost under four different estimates of the Wasserstein gradient. We will make more clear this and further expand the discussion on the speed/computational order in the main text.
>
> Section 3.4 clarified that, for our application, we employ the functional gradient used in Stein Newton Method (with diagonal Hessian approximation) in WGBoost at the end rather than one used in SVGD. It is the second-order version of the functional gradient in SVGD and more efficient. Our simulation study shows that this choice has better loss-decay and computation time.

---

> ### Comment · Reviewer_q9Jy · 2024-08-13
> **Thank you for the reply**
>
> Thank you for your response to my questions.
>
> However, I still have concerns. Specifically, the fundamental and mathematical differences between SVGD and your proposed methods remain unclear. Although you mention that the purpose of the methods differ—sampling versus boosting—the core principle in both cases involves minimizing the objective functional, such as KL, L2, within an RKHS framework [1, 2]. This suggests that, despite the different objectives, the algorithms and their mathematical foundations might be essentially similar. Therefore, I would like to keep the score as is.
>
> [1] Stein Variational Gradient Descent as Gradient Flow
> [2] On the geometry of Stein variational gradient descent

---

> ### Author Response · Authors · 2024-08-13
> **Response to Reviewer**
>
> Thank you for your response and the opportunity to clarify.
>
> > the algorithms and their mathematical foundations might be essentially similar (to SVGD)
>
> Please let us make our last attempt to clarify the difference. Here we will compare them from the viewpoint of sampling. While SVGD samples from one distribution, WGBoost samples from a **conditional** distribution as follows:
>
> - **SVGD** gets samples $(\theta_1, \dots, \theta_N)$ from a distribution $P(\theta)$, where let's say $P(\theta)$ is an arbitrary distribution over an arbitrary variable $\theta$.
> - **WGBoost** gets samples $(\theta_1(x), \dots, \theta_N(x))$ from a conditional distribution $P(\theta \mid x)$ for any input $x$. WGBoost learns the conditional distribution in the setting where we can know the form of $P(\theta \mid x_i)$ only at given data inputs $( x_i )\_{i=1}^{n}$. In the algorithm, WGBoost **predicts** the Wasserstein gradient used to sample $(\theta_1(x), \dots, \theta_N(x))$ for each input $x$ by tree-model.
>
> SVGD do not have input variable $x$ so do not have the prediction stage of the Wasserstein gradient unlike WGBoost. In this sense, WGBoost is a conditional version $P(\theta \mid x)$ of a Wasserstein gradient flow of $P(\theta)$, where the Wasserstein gradient of the particles $(\theta_1(x), \dots, \theta_N(x))$ will be predicted for each input $x$ because the form of $P(\theta \mid x)$ can be known only at $x=x_i$ in our setting.
>
> I hope this way of explanation concisely shows the difference. We are keen to hear if the reviewer's score remains unchanged.

---

### Official Review · Reviewer_UM7J · 2024-07-12

**Soundness:** 2
**Presentation:** 2
**Contribution:** 2
**Rating:** 3
**Confidence:** 3

**Summary:**

This paper introduces a probabilistic boosting tree algorithm called Wasserstein boosting that uses a smoothed particle gradient to provide probabilistic predictions. Experiments are performed using UCI tabular regression, and out of distribution classification.

**Strengths:**

Originality:

-	I like the application of Wasserstein gradients and gradient flow to gradient boosting trees, don’t think I’ve really seen any paper like this before.

Quality:

-	I didn’t check especially carefully but the machinery for the algorithm seems to be well explained and correct.

Clarity:

-	The relevant machinery of Wasserstein gradients and gradient boosting is mostly well explained.

Significance:

-	Making trees more probabilistic with limited modifications to their tabular capabilities would be a quite nice advance.

**Weaknesses:**

Originality:

-	Nothing really noted. Being a straightforward application of some machinery is totally fine.

Quality:

-	For a paper on trees, I find the experiments quite small scale and limited. I would have expected NLL experiments on larger scale datasets, such as the xgboost and lightgbm papers.

-	Comparisons to (approximate) Bayesian neural networks are quite weak – many other methods such as sgmcmc, etc. tend to outperform on datasets of these scales. references: https://arxiv.org/abs/1902.03932, https://arxiv.org/abs/1907.07504, https://arxiv.org/abs/2002.03704, amongst others

-	The natural missing comparison here is to Gaussian processes and other kernel methods, which are naturally probabilistic and similarly nonparametric (like trees).

-	Another missing set of experiments is comparison to quantile regression  / pinball loss using trees, which is implemented directly in lightgbm. Quantile regression itself is also naturally nonparametric in at least some sense.

Clarity:

-	My understanding of Wasserstein particle flows is that the particles should interact in some manner during the gradient step. However, the writing of Algorithm 1 makes this quite unclear. I think that the kernel smoothing in the gradient step for the approximate flow is what makes the particles interact, but it’s overall quite unclear.

         o	The code doesn’t seem to provide any clarity here.

-	Overall, it’s quite unclear which parameters in the loss are actually being estimated. If we’re only estimating uncertainty in regression parameters (or analogously classification), then there’s straightforward two stage approaches.

      o	For example, one can easily fit a tree predicting the mean and then modify the loss function to predict its variance, or we can modify the loss in classification problems to do something analogous.



-	The writing is extremely passive and non-specific. Suggestions below.

       o	L150: “procedure of exact or approximate” Please use the algorithm box to specifically write or point out which algorithm is used in the experiments. The current algorithm is so unspecific it’s very hard to follow.
       o	L113-115: “Although [32] … originally suggested…” rephrase to something like “Although Friedman [32]  originally proposed using a line search … , Buhlmann and Hothorn [34] recommend against the line search …”

Significance:

-	Part of the strength of trees in my experience is that they scale pretty well to large tabular datasets (e.g. n = 10 million). You also tend to need strong uncertainty quantification on these types of datasets, which is part of the reason why Bayesian neural nets became popular for a while. Yet, these large scale uncertainty quantification experiments are lacking from the paper.

      o	Bayesian neural nets: https://proceedings.mlr.press/v115/izmailov20a.html, https://proceedings.mlr.press/v130/immer21a/immer21a.pdf,

      o	Gaussian processes: https://arxiv.org/abs/1809.11165,

**Questions:**

Algorithm 1:

-	Is the Wasserstein gradient where the particles manage to interact? Otherwise, I see nowhere else the various particles / learners would interact.

Table 1: where do the error bars in the baseline methods come from?

-	Was the same preprocessing done / e.g. same train / test split for these?

-	In general, I’m pretty sure that MC Dropout is a quite weak baseline for a Bayesian neural net, and many other BNN approaches are much stronger than it at this point (see references above)

-	the clear tree baselines here would be quantile regression (implemented in lightgbm) and fitting a mean model with MSE loss and then a second tree to predict the variance by optimizing $\max s = N(y | \hat y, \exp\{s\})$ which can be done in lightgbm with a manual loss function.

Section 3.3:
-	so we’re just doing Wasserstein gradients on the mean and variance in a Gaussian regression problem? If so, then it seems like a mean / variance two stage fit model would be the natural “base” method to compare to (and could still plug in the “prior” as regularization”)

Section 3.1:

L190: “it depends only on the log gradient …”: yes, this is the requirement for most Bayesian (and probabilistic) inference, e.g. MCMC sampling, variational inference techniques.

Table 1: what are the error bars?

Table 2: I believe that MC Dropout is pretty well known for being poor at out of distribution detection [find reference], while the P Network is certainly a stronger baseline. What would entropies of a traditional multiclass classification tree perform like in terms of an out of distribution detector here?

---

> ### Author Rebuttal · Authors · 2024-08-04
>
> Thank you for your constructive feedback to our work. We respond to each of your comments and concerns below. We hope that these responses will open up room for an upward revision of the score of our work. (In our response below, citation numbers such as [9] and [10] correspond to references in our manuscript.)
>
> &nbsp;
>
> > For a paper on trees, I find the experiments quite small scale and limited ... / ... large scale uncertainty quantification experiments are lacking ...
>
> Thank you for the opportunity for us to strengthen our experiment. We will add new experiments on large scale datasets, e.g. HIGGS, KEGG, Buzz in social media, Year Prediction MSD datasets which all have very large size n>500000. Meanwhile, the current set of the experiments would be a common benchmark in UQ literatures [13, 33, 9, 10] and include medium-large datasets of size n≈50000 or n≈10000 (For example, a relevant gradient-boosted tree UQ method, Natural gradient boosting [33] in NeurIPS, used only those datasets in Section 4.2). We believe that adding the large scale experiment to the current experiments would make our benchmark strong for both the tree and UQ contexts.
>
> &nbsp;
>
> > Comparisons to (approximate) Bayesian neural networks are quite weak ...
>
> Thank you for raising the point together with the expert literatures. We will include advanced approximate Bayesian methods beyond MC Dropout in our comparison. The previous manuscript followed our relevant UQ literatures [33, 13, 14] who used MC Dropout and Deep Ensemble in comparison.
>
> Here, we have checked experimental results of the provided literature, Subspace Inference for Bayesian Deep Learning. Their Appendix Table 2&3 shows Log-Likelihood (-1 times NLL) and RMSE scores of several advanced Bayesian methods on some datasets we used. Our method produces the best score in 8 rows out of 10 rows in Table 2&3 combined.
>
> &nbsp;
>
> > The natural missing comparison here is to Gaussian processes and other kernel methods, ... Another missing set of experiments is comparison to quantile regression / pinball loss using trees, ...
>
> Thank you for your constructive feedback. We will also include kernel-based and quantile-based nonparametric methods in our experiments. With these methods added, our comparison covers a range of methods: common deep learning UQ methods, other approximate Bayesian methods, other gradient boosting UQ method, and nonparametric regression methods. We believe that this gives us a high-quality benchmark for our proposed method.
>
> &nbsp;
>
> > The code doesn’t seem to provide any clarity here / The current algorithm is so unspecific it’s very hard to follow.
>
> Thank you for the opportunity for us to clarify. First, we would like to recap our current paper structure. In Section 2, we firstly explained an abstract framework of WGBoost in Algorithm 1. To use WGBoost, we have to specify
>
> - (a) a loss functional over probability densities;
> - (b) how to estimate/approximate the Wasserstein gradient of the chosen loss.
>
> These (a) and (b) are user-specified components of WGBoost. Then, in Section 3, we presented the specific setting where we use (a) the KL divergence and (b) the kernel-smoothing estimate of the Wasserstein gradient, for our application. So it was our intention that we described the algorithm in a general manner in Algorithm 1, and then specified (a) and (b) later in Section 3.
>
> We will more strongly clarify that WGBoost in Algorithm 1 takes (a) and (b) as user-specified components. In Appendix, we have Algorithm 3 that shows the explicit algorithmic table for the specific setting in Section 3. We will clarify this as well. Finally, the terminology "exact or approximate Wassetstein gradient" would be confusing so we will change it to "estimated Wasserstein gradient".
>
>
> &nbsp;
>
> > My understanding of Wasserstein particle flows is that the particles should interact in some manner during the gradient step. / Is the Wasserstein gradient where the particles manage to interact?
>
> Thank you for checking this point with us. In principle, interaction is not compulsory part of Wasserstein particle flows. Whether particle interaction happens or not depends on how to estimate the Wasserstein gradient. If it happens, it is beneficial to have some enforced dispersion of particles.
>
> For example, the JKO-scheme-based estimation of the Wasserstein gradient does not have interaction term. Langevin Monte Carlo---a stochastic formulation of Wasserstein flow--does not have interaction term neither. On the other hand, our case of the kernel smoothing estimate in Section 3 indeed induces particle interaction. So interaction is not compulsory but the WGBoost under the setting of Section 3 for our application indeed has particle interaction thanks to the choice of the estimate.
>
> &nbsp;
>
> > Table 1: where do the error bars in the baseline methods come from? / Was the same preprocessing done / e.g. same train / test split for these?
>
> We followed a de-facto standard protocol of [53] to prepare 20 different patterns of training/test sets from the datasets in Table 1. The error bar was created by computing the test error 20 times using the 20 different patterns.
>
> &nbsp;
>
> >  it seems like a mean / variance two stage fit model would be the natural “base” method to compare to ...
>
> WGBoost produces a particle-based "distributional" estimate of the mean and variance of the Gaussian-noise regression for each input. Indeed a method that produces a point estimate of mean and variance would be a good base model, and NGBoost is such a tree method in our comparison. We will further investigate on the two stage model too.
>
> &nbsp;
>
> > What would entropies of a traditional multiclass classification tree perform like in terms of an out of distribution detector here?
>
> We didn't include this in our manuscript but observed that entropies of a traditional classifier shows poor OOD performance during the development of our method. We will add a discussion on this point.

---

> > ### Comment · Reviewer_UM7J · 2024-08-11
> >
> > I thank the authors for their clarifications. However, without further experimental comparisons, my opinions on the paper are unchanged.
> >
> > In terms of experimental benefits, I don't think there's a compelling application as described in the paper to use Wasserstein boosting for tree methods as compared to something like NGBoost (or other probabilistic baselines like 2 stage fits, quantile regression etc.). Wasserstein boosting provides non parametric uncertainty estimates, which NG Boost and quantile regression on trees also provide. I understand that Wasserstein boosting could be used on top of a quantile regressor or the two stage fitter, but these experiments just aren't done presently so there's, to my understanding, not as much of a compelling story here.
> >
> > re " , Subspace Inference for Bayesian Deep Learning. " I believe the pre-processing may be slightly different (you can see by comparing their "SGD" baseline to your deep ensemble baseline), but this is good to understand that the Wasserstein boosting algorithm performs quite sensibly.
> >
> > I also think that, even after the rebuttal, the comparison with SVGD is a bit confusing to me (as mentioned by reviwer q9Jy), and would like to see the authors further clarify here. The algorithm seems to be different primarily in its applications to boosting, but in some sense this is quite similar to the application in parametric models as well.

---

> ### Author Response · Authors · 2024-08-12
> **Response to Reviewer Comment**
>
> We appreciate the reviewer for providing further comments and opportunity for us to discuss about them. All your comments will be used and helpful to improve our manuscript.
>
> (PS: Please refresh this webpage in case the latex commands were not rendered as math equations properly.)
>
> &nbsp;
>
> ## Difference from NGBoost and Quantile Regressor
>
> To illustrate, we would like to consider a classification problem, where we have input $x$ and output label $y$. We assume a categorical distribution $C(y \mid q)$ over the label $y$ given the class probability vector $q$. What NGBoost (and two-stage fit model) will provide is a **point estimate of the class probability** $q$ at each input $x$, that is
>
> - $x \to \text{NGBoost} \to q(x) \to C(y \mid q(x))$
>
> On the other hand, what WGBoost will provide is a particle-based **distributional estimate of the class probability** $q$:
>
> - $x \to \text{WGBoost} \to ( q_1(x), \dots, q_N(x) ) \to (1 / N) \sum_{i=1}^{N} C(y \mid q_i(x))$
>
> What we show in the paper is that (i) taking the average over the distributional estimate leads to better performance (in Section 4.2 with better performance than NGBoost for regression) and (ii) uncertainty in the distributional estimate can be used for the OOD detection (in Section 4.3). These advantages are also different from quantile regressors since quantile regressors provide quantile of output variables $y$ only i.e. no distribution on $y$ nor distribution of $q$. (Such a distributional estimate of the class probability $q$ has been explored in Deep Evidential Learning (DEL) approaches. In this context, WGBoost is the first method that enables the tree version of DEL approaches.)
>
> &nbsp;
>
> ## Difference from SVGD
>
> Firstly, we would like to clarify difference in what WGBoost and SVGD can do:
>
> - **SVGD**: Given one target density $\pi$, we can obtain samples $(\theta_1, \dots, \theta_N)$ form $\pi$.
> - **WGBoost**: Given a set of several inputs and output-densities $( x_i, \pi\_{x_i} )\_{i=1}^{n}$, we can obtain **a map from input $x$ to particles $(\theta_1(x), \dots, \theta_N(x) )$** that approximates the given $\pi\_{x_i}$ for in-sample input $x_i$ and also predicts an unseen $\pi\_{x}$ for new out-of-sample input $x$.
>
> Next, we would like to clarify how WGBoost and SVGD is applied for clarification/regression. Let us consider the classification again, where we have input/label dataset $(x_i, y_i)\_{i=1}^{n}$:
>
> - **SVGD**: We have a model $f(x; w)$ with the parameter $w$ that outputs the class probability $q$. We use a **posterior $Pos(w \mid ( x_i, y_i )\_{i=1}^{n} )$ of the model $f(x; w)$ using all data points**. SVGD produces samples $(w_1, \dots w_N)$ from the posterior $Pos(w \mid ( x_i, y_i )\_{i=0}^{n} )$.
> - **WGBoost**: WGBoost is a learning model to estimate $q$ distributionally, so there is no other model $f(x; w)$ like the SVGD case. We use a **posterior $\pi\_{x_i}$ of the categorical distribution $C(y \mid q)$ for each single data point** $(x_i, y_i)$ like done in some DEL approaches:
>
>   - $\pi\_{x_i}(q) \propto C(y_i \mid q) \times \nu\_{x_i}(q)$ for each $(x_i, y_i)$, where $\nu\_{x_i}$ is a prior over $q$ given at each $(x_i, y_i)$.
>
>   WGBoost produces a map from input $x$ to particles s.t. $x \to \text{WGBoost} \to ( q_1(x), \dots, q_N(x) ) \approx \pi_x$.
>
> &nbsp;
>
> Finally, please let us elaborate the intuitive algorithmic idea of WGBoost. The challenge is how to produce particles for unseen input $x$:
>
> 1. We have the given set $( x_i, \pi\_{x_i} )\_{i=1}^{n}$. First let's sample from every given $\pi\_{x_i}$ using any user-choice of Wasserstein gradient flow (WGF) such as SVGD;
> 2. The WGF for each $\pi\_{x_i}$ uses the Wasserstein gradient $g\_{x_i}$, so we get a set of the computed Wasserstein gradients $(x_i, g\_{x_i})\_{i=1}^{n}$. Since $g\_{x_i}$ is finite-dimensional, we can train a ML model that predicts the gradient $g_x$ for new $x$ by fitting it to the set $(x_i, g\_{x_i})\_{i=1}^{n}$.
> 3. The trained ML model gives us a prediction of the Wasserstein gradient $g_x$ for unseen input $x$. So let's perform the WGF with the predicted gradient and have predictive particles even for unseen input $x$.
>
> This procedure uses a user-specified WGF (e.g. SVGD) for every $\pi\_{x_i}$ and train a ML predictive model with the computed Wasserstein gradient at $x_i$. So SVGD is simply used as an intermediate component of WGBoost. Rigorously formalising this idea requires academic work, for which we showed that it can be formalised as an extension of gradient boosting.
>
> &nbsp;
>
> ## Comment
>
> Would this clarify difference from NGBoost and SVGD? We have added these better clarifications in comparison with SVGD to the main text. To date, adapting Wasserstein gradient flows to other gradient-related ML methods is still underdeveloped. We truly believe this work can bring a new inspiration to the ML community, bridging Wasserstein gradient flows and gradient boosting for the first time. We are keen to hear any opinion from the reviewer.

---

> > ### Comment · Reviewer_UM7J · 2024-08-13
> >
> > Thanks for your further responses .
> >
> > > WGBoost is the first method that enables the tree version of DEL approaches.
> >
> > Indeed, this is a selling point (and a drawback as the inference should be more expensive..) of your approach as compared to other tree based methods from my understanding. However, it's not clear to my why a well specfied probabilistic model can't just be used for OOD, like the early deep net based approaches were - by entropy of the multi-class predictive distribution (which NGBoost can surely gather).
> >
> > > Difference between SVGD and WGBoost
> >
> > My (somewhat non-rigorous) understanding is that with WGBoost you're essentially non-parametrically estimating the integral $p(\hat y | y, \mathcal{M}) = \int p(\hat y | f(x)) p(f(x) | x) df$ where $f$ is the (non-parametric) functional form induced by the model class. SVGD ends up being quite similar, but (in its [original definition](https://arxiv.org/pdf/1608.04471)) operates on a _parametric_ form of model class, estimating $p(\hat y | y, \mathcal{M}) = \int p(\hat y | f_\theta(x) ) p(\theta | x) d\theta \approx p(\hat y | f_\theta(x) ) q(\theta) d\theta$, where $q(\theta) $ is the approximation distribution learnt by SVGD. However, as SVGD is a gradient flow [paper](https://arxiv.org/abs/1704.07520), it is also possible to express as a nonparametric algorithm in my understanding if we express it in terms of function space rather than parameter space.
> >
> > I guess practically they end up being different algorithms, but the underlying theory is quite similar, if not exactly the same. And there's several different function space algorithms that use stein flows as the other reviewer points out - training either neural networks or GPs with gradient flows like this is fairly well studied.

---

> ### Author Response · Authors · 2024-08-13
> **Response to Reviewer**
>
> Thank you for your response and the additional opportunity to discuss.
>
> > However, it's not clear to my why a well specfied probabilistic model can't just be used for OOD, like the early deep net based approaches were - by entropy of the multi-class predictive distribution (which NGBoost can surely gather).
>
> Thank you for the suggestion. As in the previous comment, we observed that entropy of multi-class classifier didn't perform well and decided to focus only on the uncertainty done in the current paper for the page limit. We will include entropy of multi-class classifier in the comparison. In our understanding, our experiment has already shown better performance of WGBoost on small to medium-large datasets, so adding large-scale dataset and other comparison would complement our experiment at sufficient level.
>
>
> > underlying theory is quite similar (with SVGD)
>
> Thank you for the discussion about this point. Infinite-dimensional SVGD has been studied and the algorithm/theory is different [1]. Please let us attempt to clarify the difference again. What WGBoost learn is a conditional distribution $P(y | x) = \int P(y | q) P(q | x) dq$, where let's say $P(y | q)$ is a categorical distribution and $P(q | x)$ is a conditional distribution of the class probability vector $q$ given $x$. In this view, WGBoost is a sampler from a conditional distribution with tree-model incorporated in the procedure as follows:
>
> - **SVGD**: Let's say $P(\theta)$ is an arbitrary distribution over an arbitrary variable $\theta$. SVGD gets samples $(\theta_1, \dots, \theta_N)$ from a distribution $P(\theta)$ (regardless of $\theta$ is finite-dimensional or replaced with infinite-dimensional variable $f$);
> - **WGBoost** gets samples $(\theta_1(x), \dots, \theta_N(x))$ from a conditional distribution $P(\theta \mid x)$ for any input $x$. WGBoost learns the conditional distribution in the setting where we can know the form of $P(\theta \mid x_i)$ only at given data inputs $( x_i )\_{i=1}^{n}$. In the algorithm, WGBoost **predicts** the Wasserstein gradient used to sample $(\theta_1(x), \dots, \theta_N(x))$ for each input $x$ by tree-model.
>
> SVGD do not have input variable $x$ so do not have the prediction of the Wasserstein gradient unlike WGBoost. In this sense, WGBoost is a conditional version $P(\theta \mid x)$ of a Wasserstein gradient flow of $P(\theta)$, where the Wasserstein gradient of the particles $(\theta_1(x), \dots, \theta_N(x))$ will be predicted for each input $x$ (because the form of $P(\theta \mid x)$ can be known only at $x=x_i$).
>
> We hope this way of explanation clarifies the difference better? We are keen to hear if the reviewers opinion still remains unchanged.
>
> [1] Stein variational gradient descent on infinite-dimensional space and applications to statistical inverse problems

---

### Official Review · Reviewer_jx13 · 2024-07-15

**Soundness:** 4
**Presentation:** 4
**Contribution:** 4
**Rating:** 8
**Confidence:** 3

**Summary:**

The paper introduces a novel gradient boosting framework called Wasserstein Gradient Boosting (WGBoost). Unlike traditional gradient boosting methods that fit base learners to the gradient of the loss function, WGBoost fits them to the Wasserstein gradient of a loss functional defined over probability distributions. This approach is particularly useful for probabilistic prediction, where the goal is to approximate the uncertainty in the model prediction. The authors provide a general formulation of WGBoost, its algorithmic implementation, and empirical evaluations on various benchmarks.

Paper's major contributions include, the introduction of the Wasserstein gradient flow framework into gradient boosting; the development of an approximate algorithm for posterior regression using the KL divergence; and the demonstration of WGBoost's performance on regression, classification, and out-of-distribution (OOD) detection tasks. Authors also propose a second-order WGBoost algorithm built on the approximate Wasserstein gradient and Hessian of the KL divergence.

**Strengths:**

S1: The application of Wasserstein gradient flows to gradient boosting is a novel and promising direction, providing a new perspective on ensemble learning algorithms.

S2: Solid theoretical concepts, with detailed derivations and explanations of the Wasserstein gradient and Hessian approximations.

**Weaknesses:**

Authors did not specify details about the hyperparameter selection for Conditional Density Estimation, Classification and OOD Detection tasks.

Misc:
There is a typo in the line no 284, root is written as room.

**Questions:**

I would be curious to know about the scalability and the computational complexity of the proposed algorithm.

---

> ### Author Rebuttal · Authors · 2024-08-03
>
> Thank you for your strong endorsement of our work. We also believe that WGBoost is a promising direction that connects Wasserstein dynamics and gradient boosting ensemble for the first time.
>
> &nbsp;
>
> > Authors did not specify details about the hyperparameter selection for Conditional Density Estimation, Classification and OOD Detection tasks.
>
> Thank you for raising this point. We will add a clarification about the hyperparameter selection for the experiments other than Section 4.2.
>
> &nbsp;
>
> > Misc: There is a typo in the line no 284, root is written as room.
>
> Thank you for pointing this out. We have corrected the typo.
>
> &nbsp;
>
> > I would be curious to know about the scalability and the computational complexity of the proposed algorithm.
>
> Thank you for this point. In Appendix E, we have a simulation study to compare the loss decay and computational time of WGBoost under four different choices of the approximate Wasserstein gradient. We will expand the discussion about the computational cost of the algorithm in the main text (e.g. adding some big O cost order).

---

> > ### Comment · Reviewer_jx13 · 2024-08-13
> >
> > Thank you for addressing my comments. I do not have further questions and I will keep my score at 8.

---

### Author Rebuttal · Authors · 2024-08-06

We would like to express our gratitudes to all the reviewers for their efforts in assessing our work. Our full rebuttal to each reviewer has been provided in each individual rebuttal area. This global rebuttal area is used for the following contents:
- (A) Brief Summary of Each Rebuttal
- (B) Concise Overview of Our Work

&nbsp;

## A. Brief Summary of Each Rebuttal

- **Reviewer jx13** & **Reviewer DY4u**: We appreciate your strong endorsement of our work. We believe that our work represents a promising advance that combines the gradient boosting framework with Wasserstein gradient systems for the first time.

- **Reviewer UM7J**: Thank you for your constructive feedback to our work. About our experiment, we will add (i) larger-scale datasets, (ii) other advanced approximate Bayesian methods, and (iii) other kernel-based/quantile-based nonparametric methods. About clarity of our algorithm (Algorithm 1), please find our full rebuttal and Section B below recapping our paper structure. In short,
  - Section 2 explains a general framework of WGBoost in Algorithm 1, where we need to specify *'how to estimate the Wasserstein gradient'* as a user-specified component;
  - Section 3 then explains that we plug the kernel-smoothing estimate as the Wasserstein gradient estimate into Algorithm 1 for our application.

- **Reviewer q9Jy**: Thank you for your assessment of our work. We are afraid that WGBoost seems misinterpreted as a sampling algorithm (such as SVGD) in raised concerns. Briefly, WGBoost is an algorithm to solve a new form of regression problem, such that, each input $x$ is arbitrary vector and each associated output is a probability distribution $\pi_x$ rather than some vector $y$. Please find our full rebuttal and Section B below recapping the high-level problem setting of WGBoost.

&nbsp;

## B. Concise Overview of Our Work

Our paper structure is mainly twofold. We proposed a general framework of WGBoost in Section 2. We then described how WGBoost can be used for probabilistic classification/regression in Section 3.

**B.1. Problem to Solve**

Firstly, the problem that the general framework of WGBoost can solve is a new form of regression problem below:

- We have a dataset $( x_i, \pi_{x_i} )\_{i=0}^{n}$ where input  $x_i$ is arbitrary vector and output $\pi_{x_i}$ is a probability distribution given at each $x_i$. Our goal is to have a learning algorithm that predicts the unseen distribution $\pi_x$ for any new input $x$.

This is a challenging problem because the target output $\pi_{x_i}$ at each input $x_i$ is an infinite-dimensional object (probability distribution) rather than some finite-dimensional vector $y_i$. WGBoost constructs a map from any input $x$ to a set of particles $ \hat{\pi}_x$ that approximates $\pi_x$:

- $x \to \text{WGBoost} \to ( \theta^1(x) , \dots, \theta^N(x) ) =: \hat{\pi}_x \approx \pi_x$.

Our novelty is in bringing the concept of Wasserstein dynamics into gradient boosting, which has never been explored.

**B.2. Gradient Boosting with Wasserstein Gradient**

Gradient boosting is a well-used ensemble method that trains multiple weak learners and combines them iteratively. In standard gradient boosting, each weak learner is trained to approximate the gradient $\nabla_{z_i} L(z_i | y_i)$ of a loss function $L(z_i | y_i)$ between the ensemble output $z_i$ and a target-output vector $y_i$ at each input $x_i$. In WGBoost, we rather use a loss functional $\mathcal{F}(\hat{\pi}\_{x_i} | \pi\_{x_i})$ between the WGBoost output and the target distribution at each input $x_i$. Then, each weak learner is trained to approximate the "Wasserstein gradient" of the loss functional at each input $x_i$. The key innovation is that the Wasserstein gradient evaluated here will be a finite-dimensional vector. Hence, each weak learner in WGBoost can be trained like usual regression algorithms, despite the original target $\pi\_{x_i}$ is an infinite-dimensional object. See also illustrative Figure 1-2 in the attached PDF.

In gradient boosting, users have to specify which loss function $L(z_i | y_i)$ to use. In WGBoost, users have to specify

- (a) which loss functional $\mathcal{F}(\hat{\pi}\_{x_i} | \pi\_{x_i})$ to use at each input $x_i$;
- (b) how to estimate/approximate the Wasserstein gradient of the chosen functional $\mathcal{F}$.

For Wasserstein gradient flows in general, it is common that the Wasserstein gradient is not analytically available (except some nice cases) and needs to be approximated (c.f. third paragraph of Section 2.1). Hence WGBoost needs to specification of item (b) too.

**B.3. Application to Classification/Regression**

Section 3 described application of WGBoost to probabilistic classification/regression, and how we specified item (a)-(b) for it. For illustration, consider classification where we have a dataset $( x_i, y_i )\_{i=1}^{N}$ of input vector $x$ and output label $y$. Assume a categorical distribution $\mathcal{C}(y \mid q)$ over the output label $y$ with the class probability vector $q$. With WGBoost, we can obtain a particle-based distributional estimate $\hat{\pi}\_{x}(q)$ of $q$ for any input $x$ (i.e. predictive uncertainty in $q$); see Figure 3 diagram in the attached PDF. To train WGBoost, we prepare the target distribution $\pi_{x_i}$ at each input $x_i$ as follows:

- Prepare a prior distribution $\nu_{x_i}(q)$ over $q$ at each input $x_i$ (c.f. Section 3.2);
- Get a posterior distribution $\pi_{x_i}(q) \propto \mathcal{C}(y_i \mid q) \times \nu_{x_i}(q)$ over $q$ from the likelihood $\mathcal{C}(y_i \mid q)$ and the prior $\nu_{x_i}(q)$ at each data point $(x_i, y_i)$.

Note that this posterior $\pi_{x_i}(q)$ is made at each data point $(x_i, y_i)$ pointwisely. This is different from a posterior of e.g. Bayesian neural network, which is defined over the network parameter $w$ using all data, $P(w \mid (x_i, y_i)\_{i=1}^{N})$. For regression, we can think of $\mathcal{N}(y \mid m, \sigma)$ instead of $\mathcal{C}(y \mid q)$.

---

> ### Author Response · Authors · 2024-08-13
> **Final Author Comment**
>
> We would like to express our gratitudes for all the discussions. While Reviewer jx13 & DY4u agree on the novelty of our work, the remaining concern of Reviewer UM7J & q9Jy is their interpretation that WGBoost is essentially same as SVGD.
>
> In our view WGBoost is clearly different from SVGD as attempted to clarify in the rebuttal, but some of our explanation made from the viewpoint of regression so far may sound differently to some reviewer e.g. expert in sampling contexts. In this comment, we wanted to summarise our last explanation about the difference of WGBoost and SVGD from the viewpoint of sampling.
>
>
> ## Difference of WGBoost and SVGD from Viewpoint of Sampling
>
> While SVGD samples from one distribution, WGBoost samples from a **conditional** distribution as follows:
>
> - **SVGD** gets samples $(\theta_1, \dots, \theta_N)$ from a distribution $P(\theta)$, where let's say $P(\theta)$ is an arbitrary distribution over an arbitrary variable $\theta$.
> - **WGBoost** gets samples $(\theta_1(x), \dots, \theta_N(x))$ from a conditional distribution $P(\theta \mid x)$ for any input $x$. WGBoost learns the conditional distribution in the setting where we can know the form of $P(\theta \mid x_i)$ only at given data inputs $( x_i )\_{i=1}^{n}$. In the algorithm, WGBoost **predicts** the Wasserstein gradient used to sample $(\theta_1(x), \dots, \theta_N(x))$ for each input $x$ by tree-model.
>
> SVGD do not have input variable $x$ so do not have the prediction stage of the Wasserstein gradient unlike WGBoost. In this sense, WGBoost is a conditional version $P(\theta \mid x)$ of a Wasserstein gradient flow of $P(\theta)$, where the Wasserstein gradient of the particles $(\theta_1(x), \dots, \theta_N(x))$ will be predicted for each input $x$ because the form of $P(\theta \mid x)$ can be known only at $x=x_i$.
>
> We hope the contrast from SVGD is clear in this explanation. As the author-reviewer discussion period comes to end, we will leave the final decision to the AC and all the reviewers. Thank you once again for all your effort.

---

### Decision · Program_Chairs · 2024-09-25

**Decision:**

Accept (poster)

**Comment:**

Despite a score discrepancy, all four reviewers agreed that the proposed approach to predict probability distributions, based on Wasserstein Gradient Flow and Boosting, is original.

I have read the manuscript myself, and I assert that it is well-written. This "new perspective on ensemble learning algorithms" (as said by Reviewer jx13) will likely inspire future developments. The experiments are modest in scale but sufficient to show that the approach is sound and applicable to several tasks (probabilistic regression, classification, and out-of-distribution detection). Moreover, following the recommendations of reviewer UM7J, the authors have committed to expanding their experiments.

I therefore enjoin the authors to carefully take into account the reviewers' comments while preparing the camera-ready version. I also suggest briefly discussing in which manner the approach might be reminiscent of the SVGD sampling method, which will help clarify the contributions, as the authors convincingly did in their answers to Reviewer q9Jy.